# Distinct gene expression signatures comparing latent tuberculosis infection with different routes of Bacillus Calmette-Guérin vaccination

Richard F. Silver [1,2] ✉, Mei Xia[3,4], Chad E. Storer [5], Jessica R. Jarvela [1,2], Michelle C. Moyer[1,2], Azra Blazevic[3,4], David A. Stoeckel[6], Erin K. Rakey [6], Jan M. Tennant[3], Johannes B. Goll[7], Richard D. Head[5] & Daniel F. Hoft [3,4,8] ✉

Tuberculosis remains an international health threat partly because of limited protection from pulmonary tuberculosis provided by standard intradermal vaccination with Bacillus of Calmette and Guérin (BCG); this may reflect the inability of intradermal vaccination to optimally induce pulmonary immunity. In contrast, respiratory *Mycobacterium tuberculosis* infection usually results in the immune-mediated bacillary containment of latent tuberculosis infection (LTBI). Here we present RNA-Seq-based assessments of systemic and pulmonary immune cells from LTBI participants and recipients of intradermal and oral BCG. LTBI individuals uniquely display ongoing immune activation and robust CD4 T cell recall responses in blood and lung. Intradermal BCG is associated with robust systemic immunity but only limited pulmonary immunity. Conversely, oral BCG induces limited systemic immunity but distinct pulmonary responses including enhanced inflammasome activation potentially associated with mucosal-associated invariant T cells. Further, IL-9 is identified as a component of systemic immunity in LTBI and intradermal BCG, and pulmonary immunity following oral BCG.

Tuberculosis (TB), caused by the airborne pathogen *Mycobacterium tuberculosis* (Mtb), is an ongoing global health threat that continues to cause over 1.5 million world-wide deaths annually. TB vaccination with the attenuated *Mycobacterium bovis* strain Bacillus of Calmette-Guérin (BCG) has been in clinical use for 100 years; however, current intradermal (ID) vaccination provides inconsistent protection against pulmonary TB[1], the most common and contagious form of disease. Efforts to improve upon BCG therefore remain an urgent TB research priority.

Natural respiratory infection with Mtb is followed by uptake of the bacilli by macrophages of the distal airways. Following intracellular growth of the organism, these cells eventually lyse to release Mtb, which then spread via local lymphatics to mediastinal lymph nodes and, eventually, into the bloodstream. In the great majority of infected individuals, hematogenous dissemination is followed by the development of protective immune responses that serve to contain, but not eliminate the organism. This outcome of latent tuberculosis infection

[1]Division of Pulmonary, Critical Care and Sleep Medicine, Case Western Reserve University School of Medicine, Cleveland, OH, USA. [2]Pulmonary and Critical Care Medicine, The Louis Stokes Cleveland Department of Veterans' Affairs Medical Center, Cleveland, OH, USA. [3]Division of Infectious Diseases, Allergy & Immunology, Saint Louis University School of Medicine, St. Louis, MO, USA. [4]Center for Vaccine Development, Saint Louis University School of Medicine, St. Louis, MO, USA. [5]Department of Genetics, Washington University School of Medicine, St. Louis, MO, USA. [6]Division of Pulmonary, Critical Care and Sleep Medicine, Saint Louis University School of Medicine, St. Louis, MO, USA. [7]Emmes Corporation, Rockville, MD, USA. [8]Department of Molecular Microbiology & Immunology Saint Louis University School of Medicine, St. Louis, MO, USA. ✉e-mail: richard.silver@case.edu; daniel.hoft@health.slu.edu

(LTBI) provides a model of "natural immunity" or "concomitant infection" following respiratory Mtb infection that may identify components of protective immunity[2].

The propensity of memory T cells to return to the site at which they were first exposed to specific antigens, termed "homing", supports the hypothesis that immunity in LTBI individuals following natural respiratory infection provides recall responses to Mtb that are optimally localized to the lung. In prior studies of bronchoalveolar lavage (BAL) cells from LTBI individuals, we have shown marked enrichment for Mtb-responsive CD4+ T cells within the lung compared to peripheral blood[3]; further, bronchoscopic challenge with Mtb antigens recruits additional Mtb-responsive cells into the airways[4]. A meta-analysis of 23 paired cohorts indicated that LTBI individuals have 79% lower risk of developing active TB after re-exposure than naive controls[5]. Likewise, in a cynomolgus macaque model of low-dose bronchoscopic Mtb infection that can produce clinically-silent infection resembling human LTBI[6], re-challenge with bar-coded organisms demonstrated substantial protection against secondary Mtb infection[7].

Current use of BCG as an intradermal (ID) vaccination administered to newborns protects young children from disseminated forms of TB such as tuberculous meningitis. However, the ability of ID BCG to protect against pulmonary tuberculosis is inconsistent and, in many settings, quite limited. These findings suggest the possibility that ID BCG induces effective systemic immunity to Mtb, but results in suboptimal localization of recall responses to the lung. In murine and guinea pig models, respiratory administration of BCG provides greater mucosal protection than subcutaneous or ID BCG[8–10]. Bronchoscopic administration of BCG to the distal airways of rhesus macaques also provided improved protection from respiratory challenge with Mtb[11]. However, respiratory vaccination using aerosolized BCG failed to provide greater protection from Mtb than ID vaccination, although intravenous (IV) BCG was uniquely able to provide sterilizing immunity in a majority of study animals[12]. Whereas widespread respiratory and IV administration of BCG to humans require further development, mucosal vaccination via oral (PO) BCG administration has a long history of clinical safety and feasibility[13]. Although its efficacy has not been formally studied, PO administration of BCG has been uniquely proven to induce Mtb-specific mucosal immunity in human participants[14].

In the current study, we utilize RNA-Seq technology to assess gene expression signatures in an unbiased fashion from immune cells of peripheral blood and airways of control participants without prior BCG vaccination or Mtb infection, LTBI individuals, and recipients of ID and PO BCG. Our findings represent a comprehensive comparison of systemic and pulmonary immunity induced by natural Mtb infection, and by systemic and mucosal BCG vaccination in humans.

## Results

An overview of study design, study groups and analyses of differentially expressed genes (DEG), are provided in Fig. 1. Further demographic and sample information are presented in Supplementary Table I. Both blood CD4+ T cells and BAL cells were studied from groups of Mtb/BCG-naïve, BCG-vaccinated and LTBI individuals. For studies of peripheral blood, we combined the pre-vaccination studies of individuals who subsequently received ID or PO BCG. Principal component analysis (PCA) of baseline and mycobacteria-stimulated blood transcriptomal results from these individuals show largely overlapping gene expression profiles for co-cultured blood CD4+ T cells and MDDC from the two pre-vaccine groups (pre-ID and pre-PO BCG) in both unstimulated and BCG-infected conditions; these findings validate our approach of combining them into one "Mtb/BCG-naïve" group (Supplementary Fig. 1). As research bronchoscopies were not performed prior to vaccination in these individuals, a separate Mtb/BCG naïve group was recruited for studies of BAL cell

immune responses; further, because lower numbers of participants potentially eligible for bronchoscopy procedures had been involved in studies of PO BCG, we included two individuals who had received both PO and ID BCG (on the same day) for inclusion in the bronchoscopy study cohort.

## LTBI is characterized by unique baseline immune activation in both blood and BAL

Our initial plans were to focus on the ΔΔ gene expression changes associated with in vitro mycobacterial re-stimulation to characterize the TB-relevant, antigen-specific memory responses present in the mycobacteria-immune groups (previously Mtb infected or BCG vaccinated) in comparison with Mtb/BCG-naïve controls. However, as our analyses progressed it became clear LTBI individuals expressed uniquely activated baseline gene expression profiles in both blood and BAL compared to all other groups.

The unique baseline gene expression profile in LTBI was explored with a focus on immune-related gene expression themes (Fig. 2). Heatmaps showing gene expression in unstimulated cells clearly demonstrate increased expression of immune-related genes in both blood and BAL cells of LTBI individuals compared to those of all other groups (Fig. 2a, b, respectively). Examples of specific immune-associated genes that displayed increased baseline expression are presented in Supplementary Fig. 2.

We subsequently used the CompBio Assertion Engine software to compare the biological concept maps generated from the baseline gene expression data in blood and BAL from LTBI individuals, as detailed in the methods section. The findings of these two datasets demonstrate highly significant correlation as compared to randomized data ($p < 0.001$). Overlapping baseline immune findings in blood and BAL are presented in a concept map (Fig. 2c), and indicate that similar ongoing in vivo immune activation is present both systemically and in the distal airways of individuals with LTBI.

## Distinct blood CD4+ T cell gene-expression profiles in LTBI and BCG-vaccinated participants

The analytic basis for identification and presentation of gene-expression "themes" is detailed in the Methods section. Figure 3 presents gene expression ΔΔ immune theme responses from peripheral blood CD4+ T cells co-cultured with autologous BCG-infected or uninfected MDDC. Enrichment for several immune themes was observed in both LTBI individuals and in recipients of ID BCG (Fig. 3a, b, respectively). In contrast, no significant immune themes were observed in recipients of PO BCG, so no corresponding figure is presented for this group. Non-immune themes were also significantly enriched in the ΔΔ gene expression responses of LTBI and ID BCG individuals, but not in recipients of PO BCG (Supplementary Fig. 3). Figure 3c displays in tabular form the mean ΔΔ gene expression changes for all genes identified within each of twelve significantly induced immune response themes in LTBI and BCG groups; the findings emphasize that only partial overlap was observed between themes enriched in LTBI individuals and following ID BCG; none of these were significantly enriched in the PO BCG group. The ΔΔ expression of the top individual genes within three representative immune themes are displayed in heatmap formats within 3c as well. Examples of BCG-induced expression of specific genes within these immune themes are also presented in Supplementary Fig. 4.

## Distinct lung-specific transcriptomal signatures in LTBI and BCG-vaccinated groups

Mtb/BCG-naïve control participants, individuals with LTBI, and participants who had received BCG in prior vaccinations studies were recruited for participation in research BAL procedures (Fig. 1b). The ID BCG group was limited to individuals who received ID BCG alone, whereas the recipients of PO BCG included six participants

### a. Outline of Study

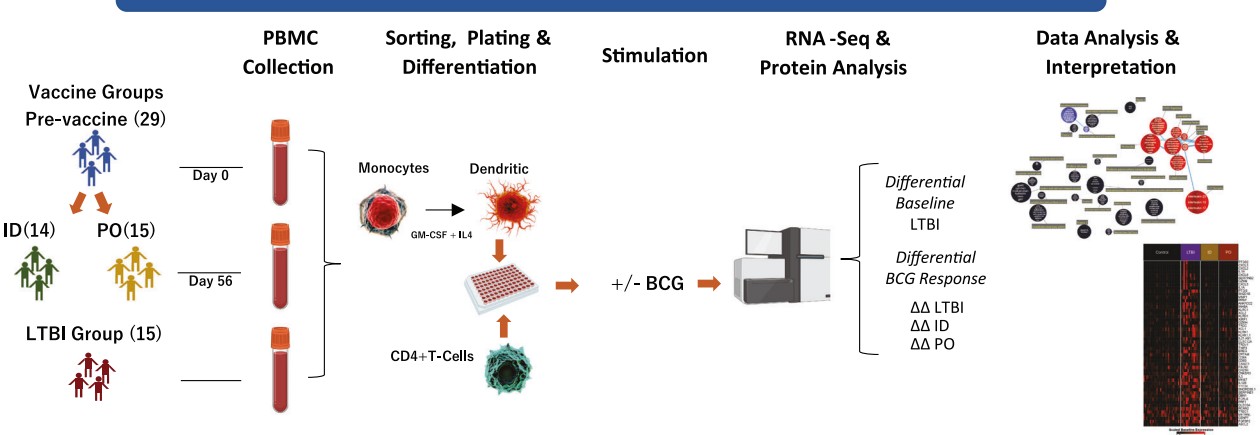

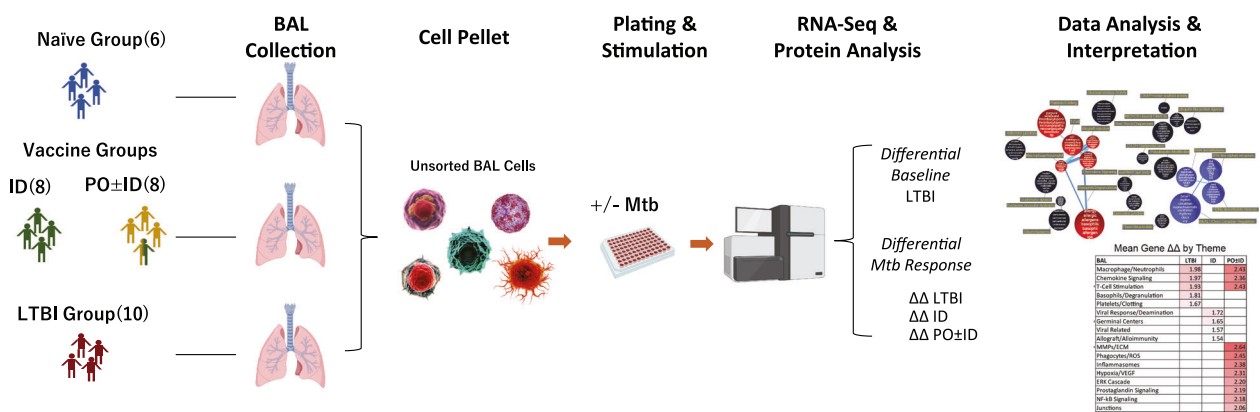

### b. Formulas

**Naive** = *Unvaccinated Individuals Without Known Mtb Exposure* **LTBI** = *Individuals with Latent Tuberculosis Infection*

**ID** = *Individuals that Received Intradermal BCG Vaccination* **PO** = *Individuals that Received Oral BCG Vaccination*

**Blood** **BAL** **Both**

$$\Delta Naive = \frac{Naive + BCG}{Naive} \qquad \Delta ID = \frac{ID_{D56} + BCG}{ID_{D56}} \qquad\qquad \Delta Naive = \frac{Naive + Mtb}{Naive} \qquad \Delta ID = \frac{ID + Mtb}{ID} \qquad\qquad \Delta\Delta = \frac{[\Delta LTBI \text{ or } \Delta ID \text{ or} \Delta PO]}{\Delta Naive}$$

$$\Delta LTBI = \frac{LTBI + BCG}{LTBI} \qquad \Delta PO = \frac{PO_{D56} + BCG}{PO_{D56}} \qquad\qquad \Delta LTBI = \frac{LTBI + Mtb}{LTBI} \qquad \Delta PO\pm ID = \frac{PO\pm ID + BCG}{PO\pm ID}$$

who received PO vaccination alone and two who received both PO and ID BCG on the same day. Demographics and BAL cell profiles are presented in Supplementary Tables Ib and Ic, respectively.

BAL cell gene expression was evaluated after overnight culture in medium alone (baseline conditions), or with virulent Mtb strain H37Rv (Fig. 4). Similar to studies of blood CD4⁺ T cells, CompBio analysis of BAL ΔΔ DEG results identified unique immune themes associated with LTBI and prior BCG vaccination; however, in marked contrast to the systemic findings, immune themes were strongly expressed in BAL cells of recipients of PO ± ID BCG as well (4a-c). Expanded CompBio ΔΔ BAL cell findings that include other non-immune themes enriched in the LTBI and BCG-vaccinated groups are

presented in Supplementary Fig. 5. Immune theme mean ΔΔ gene responses are displayed with the table in the upper-right of Fig. 4d. As displayed in this format, the PO ± ID BCG group displayed the most robust Mtb-induced immune-theme enrichment. The gene lists of specific themes are highlighted in heat maps displaying data for all participants (Fig. 4d, left and lower right). Of note, findings for the theme of T-cell stimulation indicate substantial differences in the specific enriched genes that identify this theme in LTBI individuals and in recipients of PO ± ID BCG groups, respectively. Representative examples of immune genes differentially induced by overnight Mtb infection in LTBI and both BCG-vaccinated groups are shown in Supplementary Fig. 6.

**Fig. 1 | Outline of the study: study groups, design and analytical assessment\*.** In studies of both peripheral blood CD4+ T-cell responses and of unsorted BAL cells, gene expression of LTBI individuals and of recipients of ID or PO BCG vaccination were compared to those of Mtb/BCG-naïve participants. In peripheral blood studies (Fig. 1a, top), pre-vaccination samples from participants who later received ID or PO vaccination were pooled to form the Mtb/BCG-naïve comparator group. Subsequent PBMC samples for all BCG recipients were obtained 56 days following their last vaccine dose. CD4+ T-cells were selected from thawed samples as were blood monocytes (MN) that were cultured in vitro with IL-4 and GM-CSF to develop autologous CD14-/CD11c+ monocyte-derived dendritic cells (MDDC). MDDC were then co-cultured with CD4+ T cells in the presence of BCG or in medium alone, as indicated, and samples frozen for subsequent RNA extraction. In contrast, for the BAL cell substudy (1a, lower panel), a separate group of Mtb/BCG-naïve individuals was recruited specifically as control participants; additional study groups included LTBI individuals and recipients of ID or PO BCG in prior vaccine studies. Because two of the eight individuals recruited following PO BCG vaccine also received ID BCG on the same date, we designate this group as recipients of PO ± ID BCG. Unsorted BAL cells were then incubated overnight with virulent Mtb strain H37Rv or medium alone and frozen for eventual RNA extraction. For both substudies, RNASeq was performed and data evaluated for comparisons between the study groups with regard to both baseline gene expression and differential responses to in vitro infection. \*Bacterial cell adapted from "E. coli, without flagella and pili", Blood vial adapted from "Icon Pack - Haematology", 96well plate adapted from "Icon Pack - Engineering", Sequencer adapted from "Icon Pack - Nucleic Acid Sequencing", and Lungs: Adapted from " Lung with Labels (Layout)", by BioRender.com (2023). All retrieved from https://app.biorender.com/biorender-templates. The terminology used to describe gene expression comparisons in both PBMC and BAL studies is detailed in the formulas presented in **b**. As indicated, delta (Δ) responses indicate the quotient of gene expression (in counts per million reads, or CPM) between infected and uninfected cells of a single study group in blood or BAL studies. Delta-delta (ΔΔ) responses indicate the quotient of the in vitro infection vs. uninfected baseline (Δ) in the treatment groups (LTBI or BCG-vaccine recipients) with that of the Mtb/BCG-naïve controls (Δ). Protein coding genes with a median expression ≥1.0 CPM were evaluated. For (Δ) and (ΔΔ) calculations, expression data were floored to a minimum value of 0.5 CPM. Absolute fold changes ≥1.3 with $p$-value < 0.05 (Mann-Whitney two-tailed) were required for inclusion in subsequent analysis.

## Protein-level confirmation of T cell cytokine responses in blood and BAL from LTBI and BCG-vaccinated groups

To confirm key aspects of the blood and BAL gene expression findings, we assessed BCG/Mtb-induced production of T cell-associated cytokines within supernatants from blood CD4+ T cell and BAL cultures (Fig. 5). In addition to CBA-based assessment of Th1 cell cytokines associated with protection against Mtb (IFNγ, TNF and IL-2), single-cytokine ELISA for IL-9 and IL-15 were performed due to their unexpected appearance in transcriptomic findings. In this graphic representation of data from individual participants, "ΔΔ" comparisons of statistical significance are based on the Mann-Whitney test. No IL-15 protein was detectable in any of the participant samples, so these results are not presented. Medians and ranges of all conditions and study groups, as well as additional analysis via Wilcoxon and ANOVA/Kruskal-Wallis post-hoc tests are presented in Supplementary Table II.

As detailed in the figure legends, protein-based assessments displayed good correlation with the gene-expression findings. In studies of peripheral blood CD4+ T cells, BCG-stimulation induced detectable system immunity in both LTBI and ID BCG participants, but not in recipients of PO BCG. More specifically, samples from LTBI individuals displayed significant BCG-induced production of IFN-γ, TNF, IL-2, and IL-9. Cultured CD4+ T cells from recipients of ID BCG did not produce a significant increase in IFN-γ in response to BCG stimulation, although TNF, IL-2 and IL-9 were all significantly induced. None of these cytokines were significantly induced in cultured blood CD4+ T cells from recipients of PO BCG.

Protein assessments of Mtb stimulation of BAL samples likewise were consistent with gene expression findings in demonstrating memory immune response in samples from LTBI and PO ± ID BCG groups, but not in recipients of ID BCG alone. Specifically, Mtb infection induced significant production of IFN-γ, TNF and IL-2 in cells from LTBI individuals, whereas BAL cells from the ID BCG group displayed no significant Mtb-stimulated responses for any of these cytokines. In this statistical assessment, BAL cells from recipients of PO ± ID BCG displayed significant Mtb-induced production of TNF and IL-9; however, compared to baseline for this group ("Δ" comparison), IFN-γ and IL-2 responses were significant as well (Supplementary Table II). Mtb-induced IL-9 production by BAL cells from recipients of PO ± ID BCG represented the only group in which a significant increase in IL-9 over baseline was consistently observed, although this finding must be interpreted with caution given the very low concentrations of IL-9 detected.

## Potential mechanisms of PO BCG vaccine-induced acquired immunity?

An unexpected result of our BAL gene expression analysis was the prominence of Mtb-induced expression of immune themes typically associated with innate immunity, including an inflammasome theme uniquely observed in recipients of PO ± ID BCG. Further data mining was therefore undertaken to clarify the basis for this phenomenon (Fig. 6). Presentations of ΔΔ gene expression results for an extended panel of inflammasome-related genes comparing LTBI and BCG vaccinated groups with negative controls are shown in Fig. 6a. As illustrated, the most robust Mtb-induced up-regulation of genes within this theme was observed in PO ± ID BCG recipients. Further investigations of this finding were based on consideration of key components of the inflammasome pathway (6b). Based on the critical contributions of NRLP3, IL1A and IL1B to the inflammasome pathway, we defined a "Δ inflammasome profile" constructed from the mean of the scaled Δ expression values of these three key inflammasome genes (6c). The profile was then used to assess correlations between inflammasome activation and other immune responses. The BAL scaled Δ expression values for NRLP3, IL1A and IL1B individually (red, green and blue open circles, respectively) and as a scaled combined score (connected by the red line) for each participant are presented in Fig. 6c. Although high inflammasome profile scores were observed in some individuals from each group, recipients of PO ± ID BCG displayed the most consistent elevation and, accordingly, represented the only group for which the mean score was significantly greater than that of Mtb/BCG naïve controls. Nevertheless, the inflammasome profile demonstrated strong correlations with the expression of downstream inflammasome effector molecules IL6 and CXCL8 across all study groups (6d), confirming that downstream inflammatory responses were activated.

To evaluate which processes might underlie the Mtb-induced inflammasome response, noted most prominently in recipients of PO ± ID BCG, a Spearman correlation analysis was utilized to identify genes that at baseline (prior to in vitro stimulation with Mtb infection) correlated with the Δ inflammasome profile. The resulting list of genes (all r > 0.74, p < /=5E-29) was subjected to CompBio analysis to identify relevant biological themes that demonstrated correlation with the Δ inflammasome profile (Fig. 7a, top). Interestingly, robust themes related to mucosal-associated invariant T (MAIT) cells and endosomes were strongly correlated with the Δ inflammasome profile. The expression of genes associated with this theme at baseline were most strongly observed in recipients of PO ± ID BCG, as illustrated in the

## a. Blood Baseline Expression

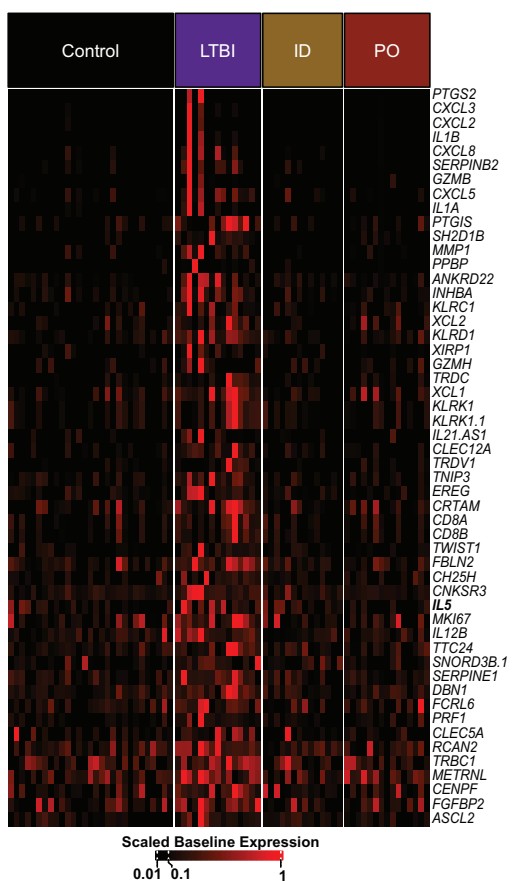

## b. BAL Baseline Expression

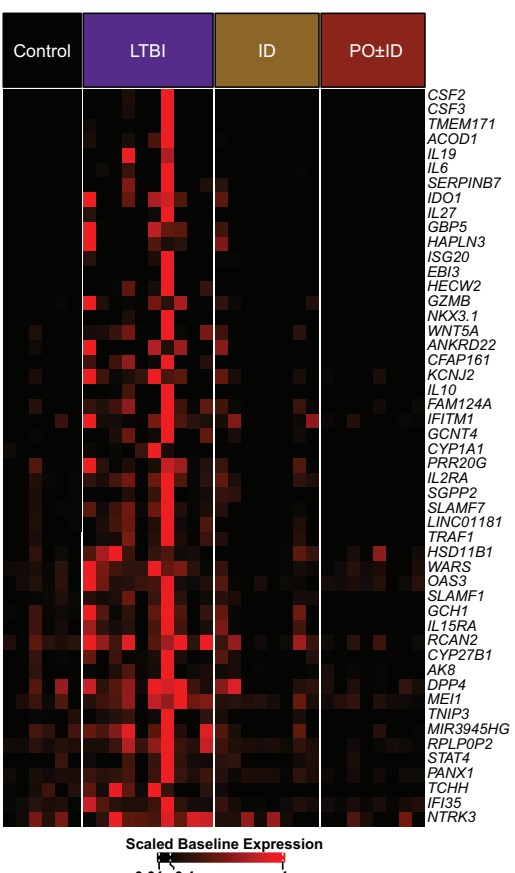

## c. Contextual Conservation of Biological Terms: Blood and BAL Baseline

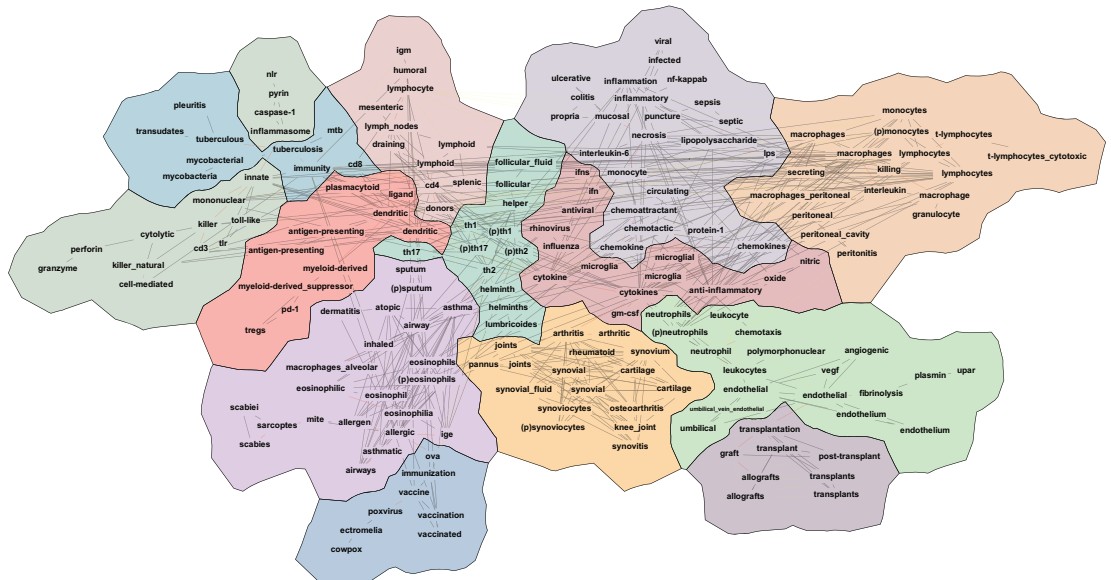

associated heat map (Fig. 7a, bottom). A strong correlation was observed across all study groups between the baseline expression of genes involved in the antigen presentation requirements for activation of MAIT cells including *MR1* (the restriction element for MAIT cell activation), and genes associated with MAIT cell-induced intracellular protective effects (phagophore formation and xenophagy induction in infected macrophages) including *GABARAPL2* (Fig. 7b, left). Further,

baseline expression of *MR1* and *GABARPL2* were significantly higher in recipients of PO ± ID BCG than in all other study groups (Fig. 7b, right). Confirmation of Mtb-induced expression of the MAIT-selective marker *DPP4* (*CD26*) in BAL further supported the presence of an increase in activated MAIT cells within the distal airways of all study groups (Supplementary Fig. 7). These findings suggest a mechanistic hypothesis in which Mtb-induced inflammasome activation may be

**Fig. 2 | LTBI individuals uniquely display ongoing expression of immune-associated genes in unstimulated cells from both peripheral blood and BAL.** Heatmaps of expression of immune response genes from ex-vivo blood CD4$^+$ T cells of LTBI individuals (in culture with autologous MDDC) demonstrate a distinctly activated profile in the absence of in vitro BCG stimulation (**a**). Parallel heatmaps of gene expression in unstimulated BAL cells are presented as well (**b**). Again, ongoing immune activation is uniquely observed in LTBI individuals and not in the recipients of BCG vaccination, nor in the dedicated Mtb-BCG naïve control group. Examples of expression of individual genes from these comparisons are presented in Supplementary Fig. 2. CompBio was used to define biologic concepts (pathways, processes, cells) associated with immune-response genes that displayed increased expression at baseline in both blood and BAL cells of LTBI individuals as compared to all other study groups. Each cohort was analyzed individually, and the resulting concept maps were then compared to each other using the CompBio Assertion Engine (Percayai); this composite map (**c**) represents concepts common to both sources of cells. The overall similarity in concepts and their inter-connectivity in the blood and BAL assessments as compared to randomized data was highly significant ($p < 0.001$). The map presents the interconnectedness of concepts with each other (indicated by connecting lines) and the grouping of strongly related concepts into map "regions" represented by distinct shading. Key concepts that appear to be strongly preserved across the baseline gene expression profiles of blood and BAL in LTBI individuals included host responses to pathogens (including Mtb), adaptive immunity including Th1 and Th17 CD4$^+$ T cell responses, inflammation, phagocytosis, and chemotaxis. In particular, among the most central and connected concepts are "cytokines" generally and "interferons" specifically, as illustrated.

mediated by memory MAIT cells. As presented left-to-right in Fig. 7c, the observed elevation in baseline expression of *MR1* and *GABARAPL2* within BAL cells of recipients of PO ± ID BCG provides a means for more efficient xenophagy of Mtb and presentation of its riboflavin metabolites to MAIT cells by their specific restriction element, *MR1*. The observed Mtb-induced expression of *IL17R*, *CD44* and *GZMB* genes, as well as production of secreted TNF protein, are all responses that may be associated with MAIT cell activation. The downstream effects of TNF can induce inflammasome-associated genes *NLRP3*, *CASP1*, *CASP5*, *IL6*, *CXCL2*, and *CXCL8*, which were all most strongly expressed in recipients of PO ± ID BCG.

## Discussion

Despite nearly 100 years of widespread ID BCG vaccination, Mtb remains one of the world's most deadly pathogens. Studies of mucosal Mtb immunity have focused attention on the route of infection or vaccination as a potentially critical factor in optimizing localization of protective responses to the lung. We present here a comprehensive evaluation of human systemic and pulmonary immune responses in LTBI individuals and recipients of standard ID and mucosal (PO) BCG vaccination. Our findings support the hypothesis that BAL findings in LTBI may serve as a model for development of protective mucosal immunity, and also demonstrate the finding that LTBI individuals uniquely display ongoing activation of both systemic and pulmonary immunity. Our studies also provide a comprehensive demonstration that despite inducing strong systemic immunity, ID BCG fails to induce optimal mucosal immunity within the lung. In contrast, the local pulmonary gene expression signature induced by PO ± ID BCG is more robust than that of ID vaccination and includes prominent Mtb-induced expression of a unique inflammasome theme. These key findings are summarized in Table 1.

Of note, the focus on comparison of findings from peripheral blood and the distal airways reported here does come with some caveats, particularly with regards to the varied methodology used in studies of blood and BAL cells. These differences reflect both differing primary interests of our two labs, as well as the practicalities of studying BAL as compared to peripheral blood. The Hoft lab has extensively studied peripheral blood responses of participants vaccinated with BCG for research purposes; in this context, a focus on CD4$^+$ T cells reflected extensive evidence indicating that this population provides critical elements of acquired immunity to Mtb; further, use of BCG for in vitro assays best reflected responses to the vaccine and were potentially adaptable to BCG vaccine trials performed in a variety of settings, including many in which lack of BSL-3 containment would exclude studies of live Mtb. In contrast, the Silver lab has focused on local immunity to Mtb as manifest within the lungs of individuals with LTBI. Given that lymphocytes generally, and CD4$^+$ T cells in particular, comprise a very small fraction of the cells in BAL, isolation of these cells is impractical for most uses; accordingly, studies of the impact of specific effector cell populations in BAL is best accomplished through depleting these populations, as Dr. Silver's lab has done[15]. The specific

methodologies used in the current project allow for comparisons of the findings with prior studies performed by each of our groups. The design adds complexity to the comparison of the two sets of studies, but also allowed for detection of novel findings within each. Specifically, the design of our peripheral blood studies would be more likely to demonstrate a wide range of CD4$^+$ effector functions given specific sorting for these cells as well as the high effector to APC ratio (20:1) and BCG MOI (also 20:1). In contrast, alveolar macrophages (AM) are the predominant cell in unsorted BAL, resulting in effector to APC ratios in a 1:10 to 1:20 range, and Mtb infections of these samples utilized an MOI of 3:1. This design may have allowed for the unexpected finding of systemic IL-9 gene expression and protein production by both LTBI individuals and recipients of ID BCG vaccination in response to in vitro infection with BCG. The much lower percentage of lymphocytes in BAL (typically 5-10% of all cells in healthy individuals) likely reduced the ability to identify transcriptomal responses of low-frequency cells such as the Th9 subset. On the other hand, use of unsorted cells also allowed identification of an inflammasome theme signature in BAL cells from recipients of PO ± ID BCG; this is implicated as being initiated largely by MAIT cells, which predominantly display a CD4$^-$/CD8$^+$ phenotype in humans.

We hypothesized that development of LTBI following natural respiratory infection would result in effective localization of protective immune responses to the lung. In support of this hypothesis, we previously reported that cytokine production by resident airway CD4$^+$ T cells plays a predominant role in driving the Mtb-induced BAL gene expression signature in LTBI[15], consistent with murine findings that lung localization of CD4$^+$ T cells correlates with protection against respiratory Mtb challenge[16,17]. Our current findings demonstrate that, in ΔΔ comparisons with Mtb/BCG-naïve participants, LTBI individuals uniquely display acquired immunity to Mtb at both systemic and pulmonary mucosal sites; these individuals display ongoing expression of immune-associated genes in unstimulated cells from both sites as well. Because of elevated baseline gene expression, ΔΔ responses may actually underestimate the relative strength of the immune responses present in LTBI individuals in comparison to all BCG-vaccinated participants, although the significance of this issue with regard to protection from Mtb is unclear. Positron emission tomography (PET) scanning of non-human primates (NHP) with LTBI has suggested latency is not a quiescent state, but rather one of ongoing immune activity required for continued Mtb containment[6]. We cannot exclude the possibility of similar ongoing stimulation by viable Mtb in our LTBI participant group. Even though these individuals overwhelmingly reported completion of chemoprophylactic antibiotic treatment, it is well-documented that lack of full compliance with therapy is frequent, particularly for the 9-month course of isoniazid which was most commonly recommended in the U.S. during the period in which these individuals were diagnosed[18]. Further, although multiple studies have compared immune profiles of LTBI individuals and those with active TB[19–22], there is no reliable test or profile to distinguish MTB clearance from persistence in LTBI. However, protein-

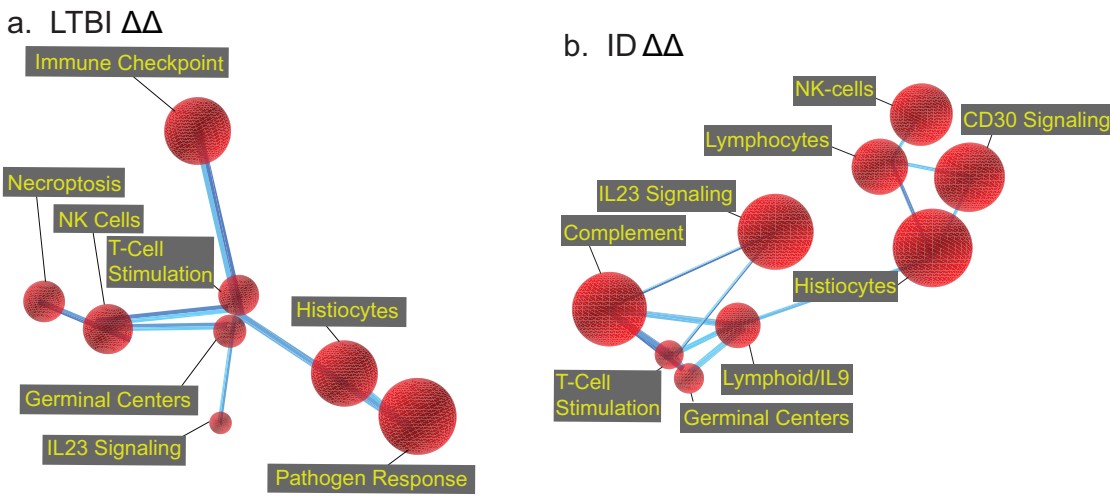

a. LTBI ΔΔ

b. ID ΔΔ

c. Immune Themes ΔΔ

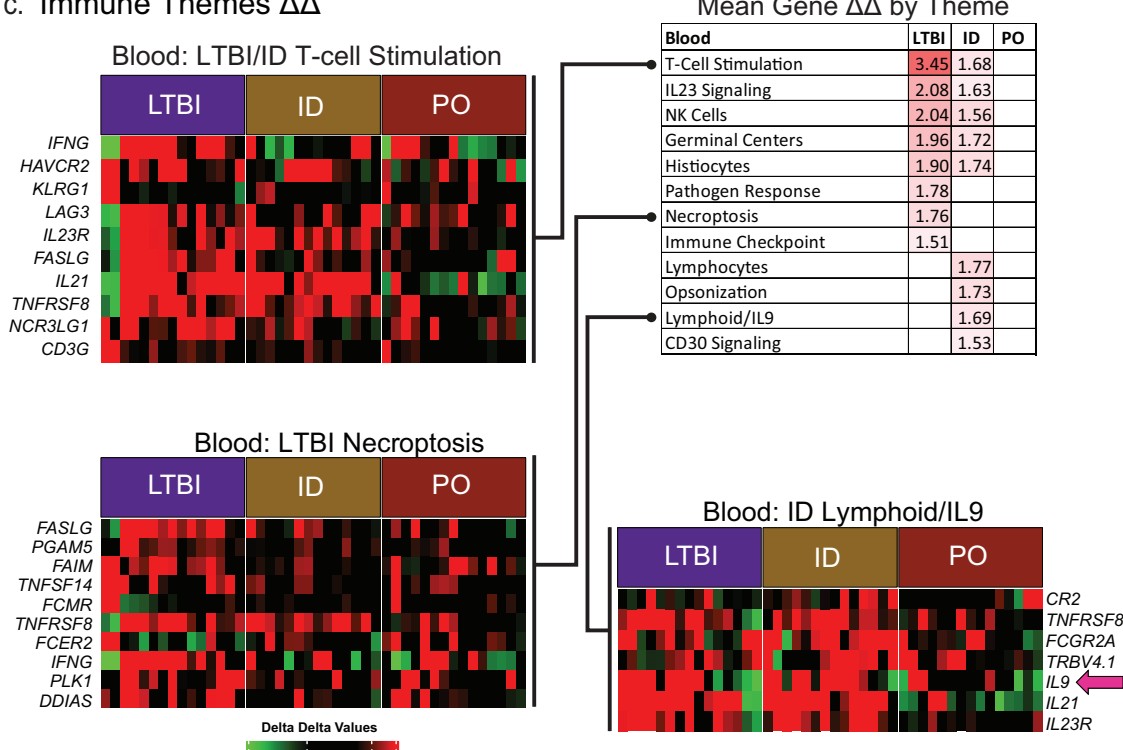

Blood: LTBI/ID T-cell Stimulation

Blood: LTBI Necroptosis

Mean Gene ΔΔ by Theme

| Blood | LTBI | ID | PO |
|---|---|---|---|
| T-Cell Stimulation | 3.45 | 1.68 | |
| IL23 Signaling | 2.08 | 1.63 | |
| NK Cells | 2.04 | 1.56 | |
| Germinal Centers | 1.96 | 1.72 | |
| Histiocytes | 1.90 | 1.74 | |
| Pathogen Response | 1.78 | | |
| Necroptosis | 1.76 | | |
| Immune Checkpoint | 1.51 | | |
| Lymphocytes | | 1.77 | |
| Opsonization | | 1.73 | |
| Lymphoid/IL9 | | 1.69 | |
| CD30 Signaling | | 1.53 | |

Blood: ID Lymphoid/IL9

Delta Delta Values

0.01    0.6    1.1    1.5

**Fig. 3 | Assessments of ΔΔ gene expression comparisons of blood CD4⁺ T cells in response to in vitro BCG stimulation identified multiple immune themes in LTBI individuals and recipients of ID, but not PO, BCG.** BCG-induced changes in global gene expression of cultured CD4⁺ T cells from LTBI, ID BCG, and PO BCG study groups ($n = 15, 14$, and 15, respectively) were compared to those of the pooled Mtb/BCG-naïve (pre-vaccination) group ($n = 29$) and displayed as models indicating the connectedness of red spheres that represent distinct immune themes. As detailed in the methods section, the size of each theme sphere is determined by the rank of that theme's absolute enrichment score within the full map, and the thickness of the connecting edges reflects the number of genes shared between themes. BCG-induced ΔΔ gene expression profiles identified multiple immune themes in CD4⁺ T cell from both LTBI individuals (**a**) and recipients of ID BCG (**b**), whereas no themes demonstrated significant enrichment in recipients of PO BCG. Figure 3c displays a table (upper right) of the mean ΔΔ responses for genes associated with immune-related themes detected in LTBI, ID BCG and PO BCG groups. As illustrated, both LTBI individuals and recipients of ID BCG displayed enrichment

for themes of T cell stimulation, IL23 signaling, NK cells, germinal centers, and histiocytes. LTBI individuals uniquely displayed significant enrichments for pathogen response, necroptosis and immune checkpoint themes, whereas themes of lymphocytes, opsonization, lymphoid/IL-9 and CD30 signaling themes were unique to recipients of ID BCG; again, no immune themes were significantly enriched in recipients of PO BCG. Visual comparisons of differential expression of individual genes within representative enriched immune themes are displayed in the associated heatmaps (**c**). These present the ΔΔ responses for each participant for the top ten genes within selected immune themes (or fewer when a theme was defined by less than ten genes) for which enrichment was shared by LTBI and ID BCG groups (T-cell stimulation), unique to LTBI individuals (necroptosis), and unique to recipients of ID BCG (lymphoid/IL9). Of note, both LTBI and ID BCG participants generally display upregulation of IL9 (pink arrow). Consistent with the other representations of these data, for all portions of the heatmap, expression of these genes is substantially less striking in recipients of PO BCG than in the other study groups.

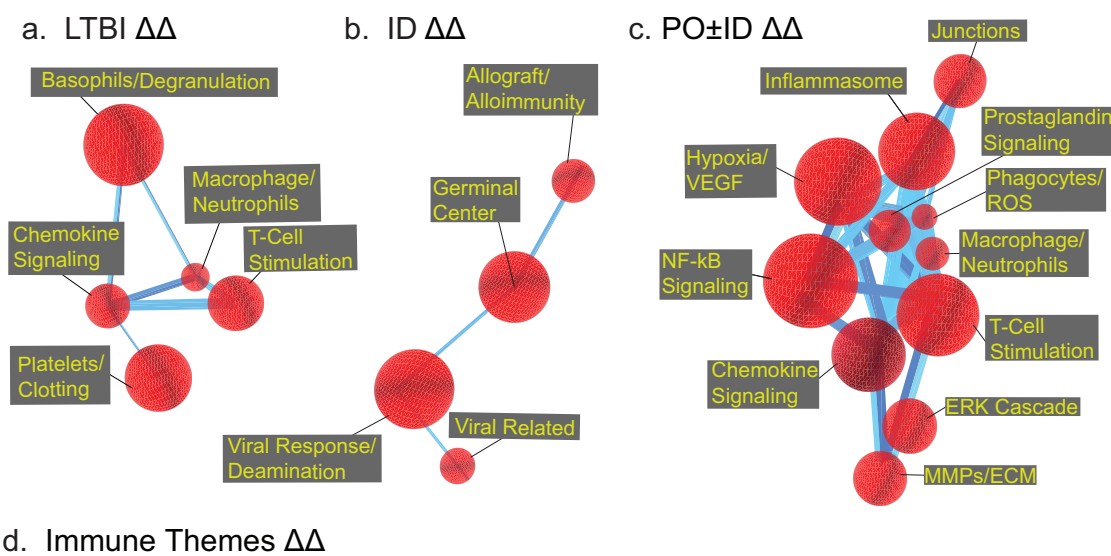

**Fig. 4 | Distinct ΔΔ gene-expression profiles of immune themes observed in BAL cells from LTBI individuals and recipients of ID and PO ± ID BCG vaccination.** Compared to control participants ($n = 6$), significant enrichments of Mtb-induced immune response themes (red spheres) were observed in BAL cells from LTBI individuals ($n = 10$, **a**) as well as in recipients of ID ($n = 8$, **b**) and PO ± ID BCG ($n = 8$, **c**). As presented in the table included with 4d (upper right), the mean ΔΔ gene expression changes for immune response genes in BAL were substantially more robust for recipients of PO ± ID BCG than for the other study groups; recipients of ID BCG alone displayed the most limited enrichment for immune themes. For the theme of T-cell stimulation, significant enrichment was seen in both LTBI and PO ± ID BCG groups; however, because the specific genes that accounted for

identification of this theme differed so greatly between the two groups, heatmaps display the top ten genes enriched in LTBI individuals (top half, upper left) followed by the top ten enriched in recipients of PO ± ID BCG (bottom half, upper left). Notably, *IL2* upregulation is strongly observed in LTBI individuals but not in recipients of PO ± ID BCG. Instead, the PO ± ID participants display upregulation of *IL15*, a potential alternative inducer of T cell proliferation (pink arrows). The additional heat maps in Fig. 4d indicate expression patterns for the top ten genes of a theme uniquely enriched in recipients of ID BCG (germinal centers, lower left heatmaps), and another uniquely expressed in those who received PO ± ID vaccination (MMPs/ECM, lower right heatmaps).

# Blood

# BAL

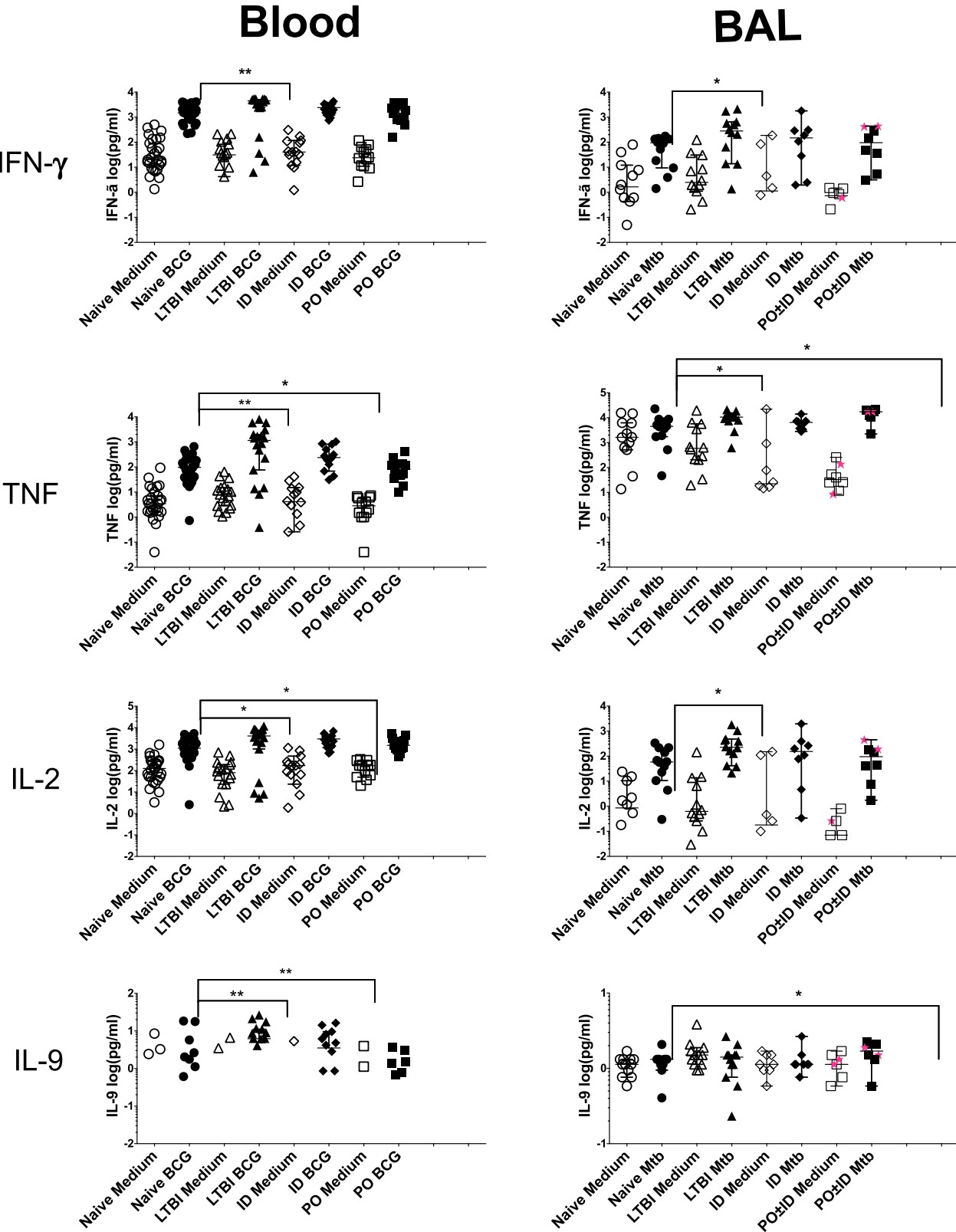

level analyses did not demonstrate ongoing production of T cell cytokines generally (or IFN-γ specifically) by unstimulated cells in blood or BAL cells in LTBI. Alternatively, these baseline findings could indicate the presence of activated or trained macrophage populations in circulation and within the lung. Hematogenous spread of Mtb during primary infection could stimulate development of these cells from bone marrow precursors as reported in mice following IV BCG[23]. In any

case, these baseline LTBI responses may provide clues for the basis of partially protective natural immunity following respiratory Mtb infection[5,7].

ID BCG vaccination induced a strong adaptive immune signature in peripheral blood, but non-significant recall responses to Mtb within the lung, consistent with the limited efficacy of ID BCG in preventing pulmonary TB[1]. Although our BAL studies were performed years after

**Fig. 5 | Protein-level confirmation of key cytokine responses significantly upregulated in the blood and BAL gene expression data.** To confirm key gene expression findings, we assessed BCG/Mtb-induced production of T cell-associated cytokines within supernatants of blood and BAL cell cultures. All graphs present cytokine concentrations in pg/mL. Responses from blood CD4$^+$ T cells are shown in the left column. Results for individual participants are shown for both uninfected cultures (open symbols) and BCG-infected samples (shaded symbols). Statistical assessment compared infection-induced changes in cytokine levels of LTBI individuals (triangles, $n = 15$) and recipients of ID and PO BCG vaccination (diamonds, $n = 14$ and squares, $n = 15$, respectively) as compared with samples from pre-vaccination Mtb/BCG-naïve controls (circles, $n = 29$). Blood CD4$^+$ T cells from LTBI individuals displayed significant BCG-induced production of IFN-γ, TNF, IL-2 and IL-9. In contrast, BCG stimulation of blood CD4$^+$ T cells from recipients of ID BCG resulted in significant production of TNF, IL-2 and IL-9, but not IFN-γ; none of these cytokines were produced to significant levels in blood CD4$^+$ T cells from PO BCG recipients. BAL cell cytokine responses are presented in the right column. Again, results for individual participants for uninfected cultures (open symbols) and infected samples (shaded symbols) are displayed, here for Mtb/BCG naïve controls (circles, $n = 6$), LTBI individuals (triangles, $n = 10$) and recipients of ID BCG (diamonds, $n = 8$). For the PO ± ID BCG group ($n = 8$), results for two participants who received PO and ID BCG are displayed as pink stars in contrast to the squares representing results of recipients of PO BCG alone. Following Mtb infection, significant production of IFN-γ, TNF and IL-2 was observed in cells from LTBI individuals. In contrast, BAL from the ID BCG group displayed minimal concentrations of these cytokines at baseline or in response to Mtb. However, BAL cells from the PO ± ID BCG group demonstrated significant Mtb-induced production of TNF and IL-9. Notably, Mtb-induced production of IL-2 was not significant in this group. Overall, the BAL cytokine protein-level data support the transcriptomic findings in demonstrating a strong local immune response in BAL cells from LTBI individuals and recipients of PO ± ID BCG, but not of ID BCG alone. *$p < 0.05$ and **$p < 0.01$ by 2-tailed Mann-Whitney U tests; error bars indicate median and 95% CI values.

the initial participation of these participants in vaccination trials, the interval between vaccination and BAL assessment was far shorter than the clinically relevant interval between standard vaccination of newborns and risk of Mtb infection in adulthood. Further, in the context of our findings, the time intervals between both ID and PO ± ID BCG vaccination and participation in research BAL were less than the intervals between diagnosis/treatment and BAL in the LTBI group (as detailed in Supplementary Table I). Together, these findings suggest that inadequate respiratory protection in adulthood following ID BCG cannot be attributed solely to waning of immunity years after vaccination, but likely also reflects limited lung trafficking induced by ID compared with PO ± ID BCG vaccination. Likewise, interference from cross-reactive non-tuberculosis mycobacteria (NTM), although potentially a factor in limiting ID BCG efficacy in other settings, is unlikely to be a confounding issue in St. Louis where this environmental exposure is uncommon.

Unlike other alternatives to ID administration, PO BCG vaccination has an extensive record of feasibility and safety in humans, as demonstrated with decades of experience with PO BCG in Brazil. It must be noted, however, that the protective efficacy of PO BCG has not been assessed in large scale human studies. Prior immunologic investigations of PO BCG at SLU have demonstrated additional measures of enhanced mucosal immunity, including increased Mtb-specific IgA in nasal washes and improved localization of antigen-specific T cells to the lungs[14,24,25]. The current data demonstrate a substantially more robust Mtb-specific BAL cell immune signature following PO ± ID BCG than ID BCG. This PO ± ID signature is qualitatively different from the Th1-dominated responses in LTBI. The Mtb-induced BAL cell signature following PO ± ID BCG suggests a role for *IL15* rather than *IL2* in development of a local T cell theme, as well as inflammasome activation associated with a potentially protective role for MAIT cells in local recall responses. Our inability to confirm local production of IL-15 at the protein level could be due to a number of potential factors, including that low concentrations were present but below the threshold for detection, or that the single time-point selected for our studies was not optimal for this assessment. BAL cells from recipients of PO ± ID BCG also displayed low, but statistically significant, levels of IL-9 protein even in the absence of identification of an *IL9* gene-expression response, which could be explained by the expected low frequency of Th9 cells among BAL cell populations contributing to this gene expression signature.

Recent publications have demonstrated strong associations between inflammasome activation and induction of adaptive immunity[26–28]. Our data suggest the inflammasome signature detected in BAL cells following PO ± ID BCG could represent an important recall response as well. Enhanced inflammasome responses strongly correlated with baseline expression of MR1, the non-polymorphic class I-like molecule that presents antigen to MAIT cells. The detection of MAIT-associated effector molecules indicates a potential mechanism for amplifying macrophage inflammasome activity. MAIT cells recognize and kill Mtb-infected monocytes[29–31] and coordinate early control of Mtb growth in vivo in lungs of aerosol-challenged mice[32,33]. Because MAIT cells are enriched in gastrointestinal tissues, PO BCG may initiate MAIT activation and development of recall responses programmed for mucosal trafficking. Our peripheral blood-based studies focused on sorted CD4$^+$ T cells, and therefore could have missed a systemic contribution by (predominantly CD4$^-$) MAIT cells. The high baseline expression of the inflammasome-associated BAL cell signature in LTBI individuals makes Mtb-induced increases in expression less dramatic in this group, but nevertheless suggests that MAIT cell-associated inflammasome responses may provide an additional mechanism of protection in individuals who also display typical Th1-mediated immunity to Mtb.

The finding of IL-9 responses in peripheral blood (in LTBI individuals as well as recipients of ID BCG) and within the airways (following PO ± ID vaccination) was unexpected. There are a few reports that IL-9 responses have been associated with infection- and noninfectious-induced varieties of inflammation (eg in pleural fluid from patients with TB pleuritis and patients with inflammatory bowel disease)[34,35]. Recent translational cancer studies in murine models demonstrated that the adoptive transfer of cancer antigen-specific Th9 cells were almost as protective against aggressive melanoma disease as were Th1 cells[36,37]. Because of the findings from this study, we undertook further investigations of the ability of IL-9-expressing Th9 cells to protect against Mtb in vitro and in vivo, which have subsequently been submitted for publication (M Xia, A Blazevic, CE Eickhoff, CE Storer, RD Head, JG Liu, J Jarvela, D Stoeckel, E Rakey, J Tennant, DL Miller, RF Silver and DF Hoft; submitted). In brief, Th9 cells and IL-9 by itself inhibit Mtb/BCG replication in both human and murine monocytes in vitro, and transfer of Mtb-specific Th9 cells protect Rag$^{-/-}$ SCID animals against aerosol Mtb challenge. Therefore, despite the expected low frequency of Th9 cells in BAL, which may have prevented detection of *IL9* transcripts from this site in human studies, our subsequent findings suggest that IL-9 may be capable of providing local protection for Mtb within the lung. Further, these early murine studies imply that the actual mechanism for how Th9 and Th1 cells mediate protective TB immunity responses are fundamentally different, suggesting that induction of Th9 cells may represent an exciting new target for TB vaccines.

The contrast between our findings in LTBI individuals and BCG recipients raises several issues regarding natural and vaccine-induced immunity to Mtb. The surprising lack of efficacy of aerosolized (AE) BCG in the NHP studies of Darrah et al was accompanied by findings that distribution of both viable BCG and antigen-responsive T cells were largely confined to the lung and thoracic lymph nodes[12]. The

**a. BAL ΔΔ Inflammasome theme**

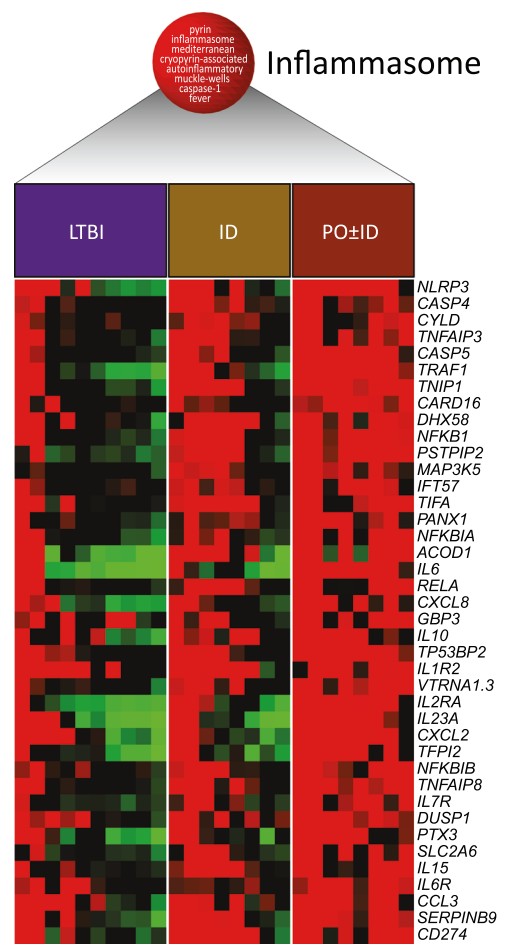

**b. Inflammasome Canonical Pathway**

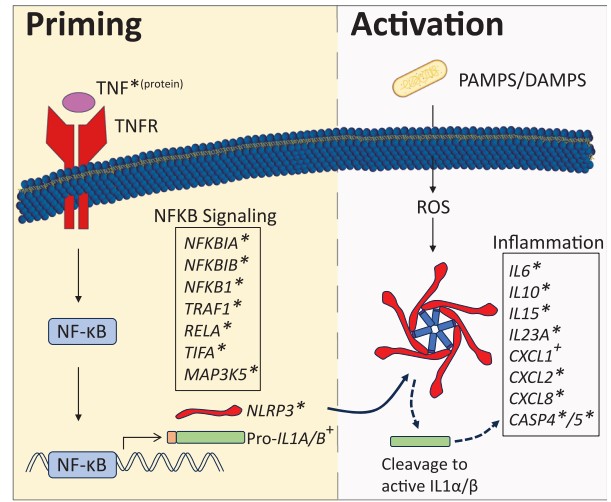

$^*$ΔΔ p ≤ 0.05 PO±ID BAL
$^+$ΔΔ p ≤ 0.10 PO±ID BAL

**c. BAL Δ Inflammasome Profile**

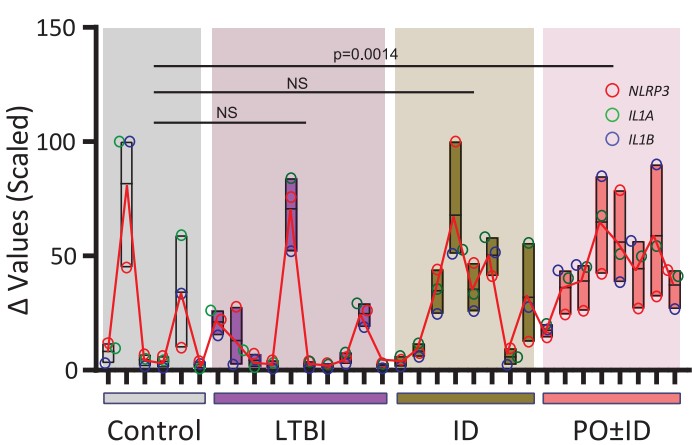

**d. BAL Δ Inflammasome-correlated Genes**

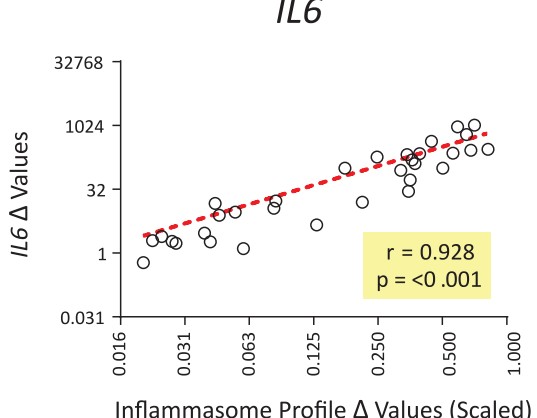

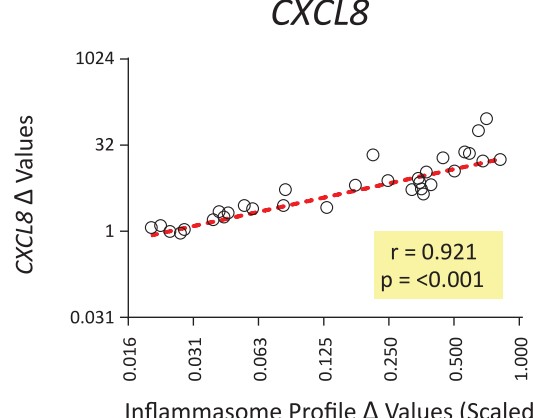

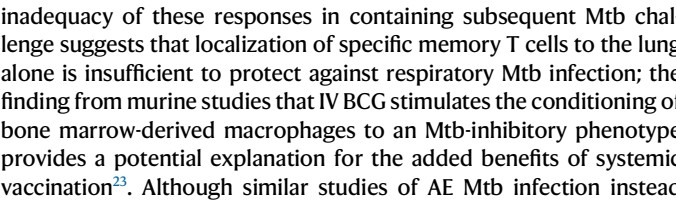

inadequacy of these responses in containing subsequent Mtb challenge suggests that localization of specific memory T cells to the lung alone is insufficient to protect against respiratory Mtb infection; the finding from murine studies that IV BCG stimulates the conditioning of bone marrow-derived macrophages to an Mtb-inhibitory phenotype provides a potential explanation for the added benefits of systemic vaccination[23]. Although similar studies of AE Mtb infection instead

induced permissive macrophages, the high bacterial burden in murine respiratory infection may fail to model the outcome of more limited Mtb dissemination in immunologically-intact humans. Indeed, a murine model of contained Mtb infection following ID inoculation of the ear was associated with enhanced protection against Mtb through ongoing innate immune activation[38]. It may be that BCG displays a more limited capacity to spread from its initial administration site than

**Fig. 6 | Mechanistic investigations of the interactions of Mtb-induced inflammasome gene expression with molecular requirements for local pulmonary immunity\*.** Heatmaps of ΔΔ expression values in BAL cells from LTBI and BCG-vaccinated cohorts for 46 inflammasome-associated genes display consistently robust responses in recipients of PO ± ID BCG (**a**). Figure 6b illustrates prominent genes involved in priming and activation phases of the canonical inflammasome pathway, indicating that ΔΔ expression for many pathway components were significantly increased (\*$p < 0.05$ and $^+p < 0.01$, respectively) in BAL from recipients of PO ± ID BCG; Mtb-induced TNF protein was observed within these samples as well (Fig. 5). TNF plays a key role in inflammasome priming through its activation of NF-KB, which induces transcription of pro-*IL1A* and pro-*IL1B*. Bacterial PAMPs activate *NLRP3* to initiate its assembly into the circular inflammasome structure. In the activation phase, further recognition of PAMPs and DAMPS induces inflammasome cleavage of pro-IL1α and pro-IL1β proteins into active forms that induce transcription of additional pro-inflammatory cytokines and chemokines. To facilitate further investigation of correlations with upstream or downstream gene expression findings, we developed an abbreviated inflammasome profile based on the expression of hallmark genes *NLRP3*, *IL1A*, and *IL1B*. Figure 6c displays Δ values

(Mtb-induced relative to unstimulated expression) of these genes in BAL cells for each participant within all four groups ($n = 6$ controls, 10 LTBI individuals, 8 ID BCG recipients, and 8 PO ± ID BCG recipients); open red, green and blue circles represent *NLRP3*, *IL1A* and *IL1B*, respectively. Boxes and lines provide ranges and means of normalized differential values for these three genes; the fine red line connects the mean scores for all participants. Across all participants, expression levels of these three genes varied similarly, providing an overall inflammasome profile value. Although observed in some individuals from each group, elevated profile scores were most consistently observed in BAL cells from PO ± ID BCG recipients. Accordingly, the PO ± ID ΔΔ inflammasome profile mean group score (scaled values of all three genes for all members of each study group) was significantly increased compared with controls ($p = 0.0014$ by Mann-Whitney, two-tailed). Spearman r value assessment indicated that the inflammasome Δ profile demonstrated strong correlation with Δ expression values of *IL6* and *CXCL8* (6d), confirming that the observed inflammasome profile was associated with downstream pro-inflammatory responses. \*Fig. 6b: Bacterial cell image adapted from "E. coli, without flagella and pili" by BioRender.com (2023), retrieved from https://app.biorender.com/biorender-templates.

Mtb, so that vaccination by any single non-IV route restricts its capacity to provide both systemic and pulmonary mucosal immunity. This possibility is supported by NHP studies of Verreck and colleagues[39]. Although their use of bronchoscopic administration of a well quantified dose of BCG did facilitate both systemic responses and protection from respiratory infection, the Mtb-derived vaccine MTBVAC provided comparable pulmonary mucosal responses but much stronger systemic immunity. As a practical consideration, future mucosal TB vaccines will be administered to the great majority of the world's population that have already received ID BCG. Aerosol administration of an adenovirus-based TB vaccine to prior recipients of ID BCG was recently shown to induce both local polyfunctional antigen-responsive T cells and conditioned macrophage populations[40]; our results suggest that PO BCG may provide an alternative means of boosting TB-specific lung immunity.

In summary, our findings illustrate that LTBI induces ongoing systemic and pulmonary signatures of Mtb-specific adaptive immunity which may serve as a model of control of Mtb and protection against respiratory re-infection. Likewise, in contrast to standard ID BCG, PO ± ID BCG is effective at inducing a localized pulmonary immune signature that suggests further potential mechanisms of protection through a lymphocyte theme dominated by *IL15* rather than *IL2*, involvement of MAIT cells in inflammasome activation, and contributions of local IL-9 production. The presence of vaccine-induced Mtb recall responses at this mucosal site could provide unique protective effects against respiratory infection and/or disease progression. The comprehensive nature of the current approach was inherently directed towards hypothesis generation; further studies in appropriate animal models and in vitro systems are therefore needed to clarify the mechanisms responsible for inducing these gene expression signatures, as well as their significance for protection against respiratory infection with Mtb.

## Methods
The studies presented in this manuscript comply with all relevant ethical regulations. Protocols for Human Subjects participation were reviewed and approved by the Institutional Review Boards of Saint Louis University, Case Western Reserve University, and the Louis Stokes Cleveland VA Medical Center. Written informed consent was obtained prior to participation in any study procedures.

### Participant groups
BCG-vaccinated individuals were participants in prior trials conducted at Saint Louis University. All BCG-vaccinated individuals were IGRA negative prior to enrollment and remained IGRA negative at the time of participation in the current study. Participants were recruited from prior

studies that had included both standard ID administration of BCG and experimental PO vaccination. All were healthy young adults with no prior history of BCG vaccination, previous close contact TB exposures, or previous known TB infection or disease. For studies of peripheral blood, all participants received BCG via only one route, either ID or PO vaccination. Because recipients of PO vaccination represented the group with the smallest target population from which we could recruit participants for research bronchoscopy procedures, we also included individuals who had received both PO and ID vaccination. Of eight recipients of PO BCG who participated in research bronchoscopy procedures, six received PO vaccination alone, and only two received both PO and ID BCG on the same day; we refer to this combined group as recipients of "PO ± ID BCG". Participants were recruited from a prior clinical trial of PO vs. ID BCG (DMID-01-351) into an RNASeq-based substudy of that trial (DMID-01-351 Transcriptomics Substudy)

LTBI individuals self-reported prior positive PPD tests or QuantiFERON TB Gold In-Tube tests and no history of BCG vaccination. Repeat PPD testing confirmed positive responses (≥10 mm induration) in all participants, and most had positive QuantiFERON results. LTBI individuals denied historical or current symptoms of TB, and chest x-rays were negative for active disease.

Mtb/BCG-naïve control participants had no prior positive PPD or QuantiFERON tests or history of BCG vaccination. All had negative confirmatory PPD tests.

Research bronchoscopy participants of all study groups were 18-50 year-old non-smokers without asthma or other chronic respiratory, cardiac, or systemic disease, and no recent systemic immunosuppression.

### Peripheral blood studies
Blood was drawn into CPT tubes and PBMC isolated, aliquoted and frozen[14]. Following thawing, CD4⁺ T cells were isolated using Miltenyi positive selection kits (Miltenyi 130-091-893). Autologous monocyte-derived dendritic cells (MDDC) were prepared as described[14]. MDDC were cultured overnight with autologous CD4⁺ T cells (20:1, T cell:MDDC ratio), in presence of either Danish BCG (Statens Serum Institut, Copenhagen, MOI 20:1), or medium alone. The next day, supernatants were harvested, frozen and stored for cytokine analyses; cell pellets were lysed for RNA extraction using RNeasy Mini Kits (Qiagen, 74106).

### BAL cell studies
Bronchoscopy with BAL was performed by instillation and recovery of up to eight 30 mL aliquots of saline[3]. Following centrifugation, cell pellets were combined, counted and aliquoted. Frozen stocks of virulent Mtb strain H37Rv (#NR-123, BEI resources, Manassas VA) were thawed, cleared of bacterial clumps, and suspended in infecting

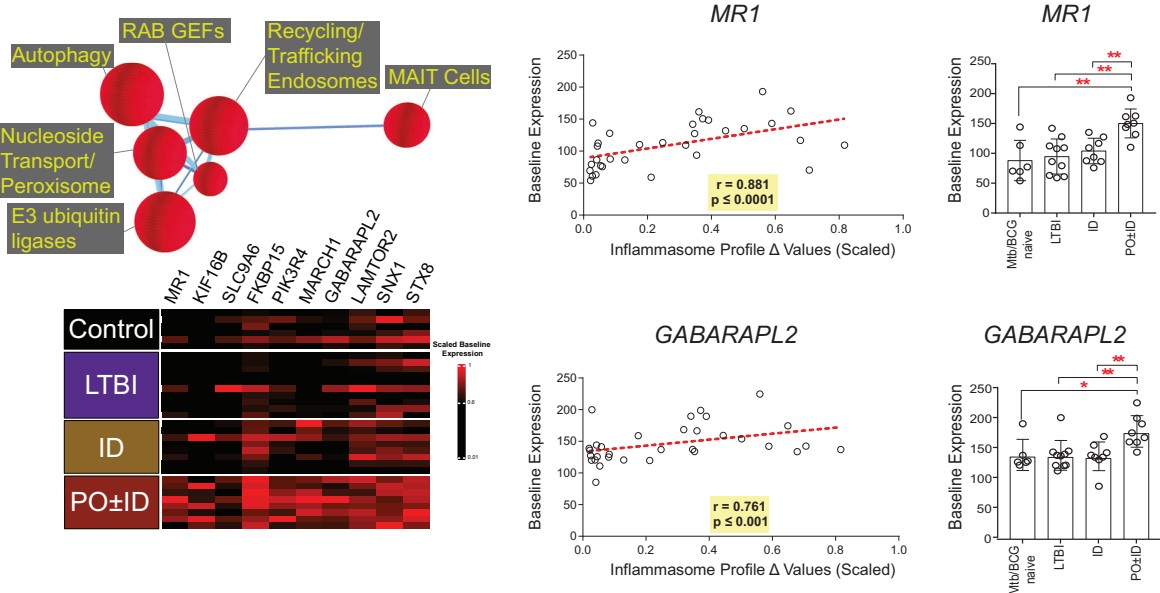

**a. Endosomal and MAIT Themes**

**b. Endosomal/MR1 Genes**

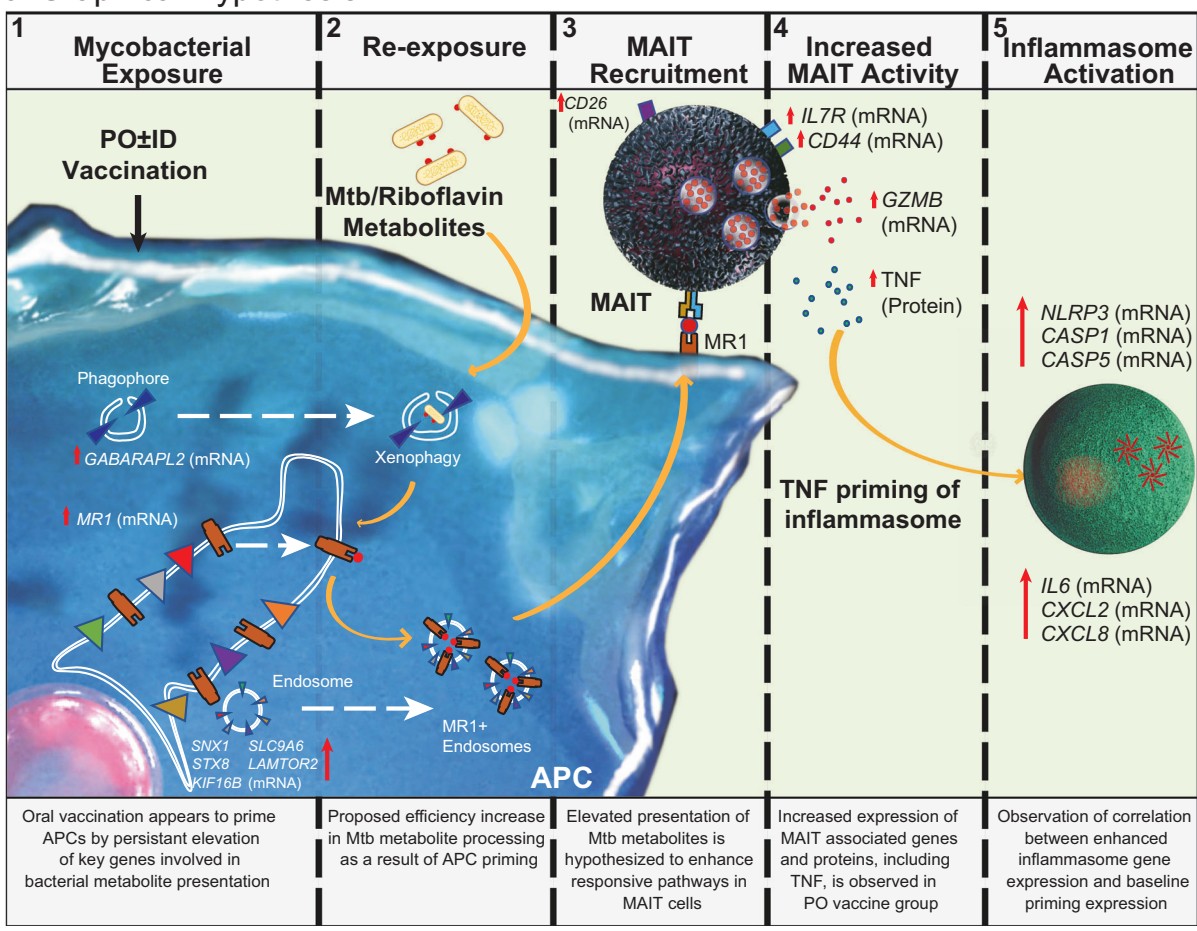

**c. Graphical Hypothesis**

medium[41]. BAL cells were cultured with H37Rv (3:1 MOI) or medium alone[15] and incubated at 37° for twenty-four hours. Supernatants were removed and frozen for subsequent use in cytokine protein assays. Cell pellets were flash-frozen and stored at −80 for eventual RNA extraction.

**RNA extraction**
Frozen cell pellets from PBMC and BAL studies were lysed using RLT buffer and RNA extracted using RNeasy Mini Kits (Qiagen, 74106). Total RNA integrity was determined using Agilent 4200 Tapestation.

**Fig. 7 | Potential mechanism of MAIT cell contribution to inflammasome activation in vaccine-induced memory responses\*.** To explore potential mechanisms behind observed differential inflammasome responses, CompBio performed a genome-wide search for genes expressed in unstimulated BAL cells that correlated with the BAL Δ inflammasome profile scores of each participant. Significantly-correlated biological themes (**a**, top) included "recycling/trafficking endosomes" and "MAIT cells". The associated heatmap displays baseline expression of ten genes associated with MAIT and endosome themes (**a**, bottom). Baseline expression of these genes was most robust in recipients of PO ± ID BCG, consistent with the greater Mtb-induced inflammasome profiles observed in these individuals (Fig. 6). Baseline BAL cell expression of *MR1* and *GABARPL2*, two genes known to be important for optimal MAIT cell responses, displayed significant correlation with the inflammasome profile across all study groups (**b**, left); expression of these genes was significantly higher in recipients of PO ± ID BCG than in any other study groups, as shown in 7b, right (\**p* < 0.05, \*\**p* < 0.01 by two-tailed Mann-Whitney test; bar graphs display mean values and SD). Again, *n* = 6 controls, 10 LTBI individuals, 8 recipients of ID BCG, and 8 recipients of PO ± ID BCG. These findings suggest a potential mechanism in which MAIT cells provide a hypothetical link between

baseline findings observed most strongly in recipients of PO ± ID BCG and subsequent inflammasome activation, **c** In steps illustrated from left to right, we hypothesize that PO ± ID BCG primes alveolar macrophages for MAIT activation through increased baseline expression of *GABARAPL2* and the *MR1* restriction molecule (step 1); these ready the APC for subsequent Mtb exposure, resulting in xenophagy that exposes Mtb-derived riboflavin metabolites and transports these to the endoplasmic reticulum to associate with MR1(step 2). MR1 endosomes then mediate transport of riboflavin /MR1 complexes to the cell surface for recognition by MAIT cells (step 3). Resulting activation of airway MAIT is suggested by Mtb-induced increases in *DPP4* (*CD26*) (also step 3). MAIT activation then triggers observed downstream inflammation including *IL17R*, *CD44* and *GZMB* gene expression and release of TNF protein (step 4). MAIT-derived TNF may then prime NRLP3 inflammasomes, providing a connection between vaccine-induced APC conditioning and Mtb-induced upregulation of inflammasome-associated genes *NLRP3*, *CASP1*, *CASP5*, *IL6*, *CXCL2*, and *CXCL8* (step 5). \*Fig. 7c: Bacterial cell images adapted from "E. coli, without flagella and pili" by BioRender.com (2023), retrieved from https://app.biorender.com/biorender-templates.

## Table 1 | Summary of key gene expression findings

| **Blood, CD4⁺** | **LTBI** | | **ID BCG** | | **PO BCG** | |
|---|---|---|---|---|---|---|
| | **Findings** | **Key Genes.** | **Findings** | **Key Genes.** | **Findings** | **Key Genes** |
| Baseline | T cell markers elevated | *MKI67* *IL9* *CXCL5* *CXCL8* | No significant immunologic findings | N/A | No significant immunologic findings | N/A |
| BCG induced (ΔΔ) | T cells; Cytokines | *IFNG* *FASLG* *IL23R* *TNFRSF8* *(CD30)* | Th9, T cells | *IL9* *TNIP3* *IL23R* *TNFRSF8* *(CD30)* | No significant immunologic findings | N/A |

| **BAL, unsorted** | **LTBI** | | **ID BCG** | | **PO ± ID BCG** | |
|---|---|---|---|---|---|---|
| | Findings | Key Genes | Findings | Key Genes | Findings | Key Genes |
| Baseline | T cells; NK cells; Elevated inflammasome markers | *GZMB* *STAT4* *DPP4 (CD26)* *NLRP3* | No significant immunologic findings | N/A | Correlation with inflammasome induction by Mtb | *MR1* *GABARAPL2* *SNX27* *KIF16B* *SCL9A6* *SNX14* *RAB7B* |
| Mtb-induced (ΔΔ) | T cells; Chemokines; Cytokines | *IL2* *CXCR2* *DPP4 (CD26)* | Modest inflammasome induction | *CD44* *DPP4 (CD26)* | T cells; Inflammasome; Chemokines; | *IL15* *IL6* *NLRP3* *CXCL8* *DPP4 (CD26)* |

As illustrated, significant observations were made from baseline studies of both peripheral blood CD4⁺ T cells from LTBI individuals (increased expression of T-cell markers) as well as unsorted BAL cells from this study group (increased expression of T cell markers, NK cells, and inflammasome markers). Following in vitro BCG infection of cultured peripheral blood cells, upregulation of T cell responses was observed in LTBI individuals and recipients of ID BCG. Whereas LTBI individuals also showed a trend for increased *IL9*, significant upregulation of Th9 responses was observed in ID BCG recipients only. In vitro infection of unsorted BAL cells with Mtb resulted in increased expression of the themes of T cells, cytokines, and chemokines in LTBI individuals, and enrichment for T cell, inflammasome, and chemokine themes in recipients of PO BCG.

## RNA-Seq Data Generation (Blood)
Library preparation used 0.5-1ug of total RNA. Ribosomal RNA was removed using Ribo-ZERO kits (Illumina-EpiCentre). mRNA was fragmented in acetate buffer by heating to 94° for 2.5 minutes, reverse transcribed using random hexamers and SuperScript III RT and underwent second strand reactions to yield ds-cDNA (Life Technologies). An A base was added to 3′ends of blunt-ended cDNA prior to ligation of Illumina sequencing adapters. Ligated fragments were amplified for 13 cycles using primers incorporating unique index tags. Fragments were sequenced on an Illumina HiSeq-3000 using single end reads extending 50 bases to achieve greater than 30 M reads per sample.

## RNA-Seq Data Generation (BAL)
Library preparation was performed with 10 ng of total RNA with a Bioanalyzer RIN score greater than 8.0. ds-cDNA was prepared using

SMARTer Ultra Low RNA kits for Illumina Sequencing (Takara-Clontech). cDNA was fragmented using a Covaris E220 sonicator. Blunt-ended cDNA had an A base added to the 3′ ends, followed by ligation of Illumina sequencing adapters. Ligated fragments were amplified for 12 cycles using primers incorporating unique dual index tags. Fragments were sequenced using Illumina NovaSeq-6000 with single end reads extending 100 bases to achieve 70 M reads per sample. All transcriptomes obtained in this study have been uploaded to Gene Expression Omnibus (GEO) for release data of November 1, 2023.

## RNA-Seq bioinformatic analysis
Differential expression analysis of RNA-seq data was performed by first excluding genes with a median expression <1.0 counts per million (CPM), and only genes defined as protein coding were examined further. Δ values were calculated as quotients of stimulated and paired unstimulated gene CPM values. ΔΔ values were calculated as quotients

of a given experimental group Δ value and the corresponding control group Δ value for each gene. For Δ and ΔΔ calculations expression data were floored to a minimum value of 0.5. Absolute fold changes >= 1.3 and p-values < 0.05 (Mann-Whitney two-tailed, non-parametric) were required for inclusion in subsequent analysis; the numbers of genes classified as significantly expressed in the various Δ and ΔΔ comparisons are presented in Supplementary Table III. Cultures of cells derived from peripheral blood were established in consistent manner across all study groups, allowing for clear interpretation of differences in gene expression across groups. In contrast, for mixed cell samples such as BAL, transcriptional data alone cannot distinguish whether the increased transcript abundance between study groups is due to upregulation of expression, difference in cellular composition with constant expression per cell, or a combination thereof. We therefore analyzed whether observed differences in gene expression showed correlation with the BAL differential cells counts (Supplementary Table I); because no correlation was found we did not include correction for these counts in our subsequent analysis.

Differential gene lists were analyzed using CompBio (V2.0 - PercayAI Inc., www.percayai.com/compbio)[42–44]. CompBio creates an extensive biological knowledge base from >33 million PubMed abstracts and >3 million full-text articles utilizing a combination of natural language processing and conditional probability analysis. The software will then simultaneously evaluate all genes in a given input list for statistical enrichment of contextually-associated biological "concepts". A CompBio knowledge map is formed when identified concepts are further assembled into biological themes, displayed graphically as spheres that represent pathways, processes, cell types, and other relevant biological mechanisms based on resulting concept-gene relationships. Themes identified by CompBio as closely associated will be proximal in the 3D landscape of the knowledge map and will share edges, often forming groups of related biology. The thickness of the edge represents the number of shared genes between the themes. The size of each theme sphere is determined by the rank of that theme's absolute enrichment score within the full map, with the first being largest. Normalized Enrichment Scores (NES) and p-values are also computed based on the absolute enrichment score and rank position. For example, Theme 1's NES and p-value are computed by comparing its absolute enrichment score to the absolute enrichment score distribution of all randomized data set Theme 1s. Given the holistic nature of CompBio analysis, stringent fold change or p-value cutoffs are not required to eliminate random noise within the input list. Significant themes (Normalized Enrichment Score > 1.3 & *p*-value < 0.1) were reviewed for biological interpretation. CompBio maps can be further compared utilizing the Assertion Engine. The Assertion Engine is a machine-learning tool that identifies not only the preservation of concepts between CompBio maps, but the preservation of their associations with other concepts within the complex biological map. The result is a 2-dimensional map which is a projection of strongly preserved concepts and relationships (shown as edges between concepts). For a concept relationship to form, not only do the shared concepts have to be in both input CompBio knowledge maps, they have to be interconnected with other concepts, multiple layers deep, with sufficient similarity to be considered a 'signal' event. P-values for global-level association signals are computed empirically through comparison of signal-containing CompBio maps to tens of thousands of randomized data sets.

### Cytokine analysis in blood CD4+ T-cell/autologous MDDC co-cultures and unsorted BAL cells

Cytokine concentrations in supernatants from BCG-stimulated blood CD4+ T cell/MDDC cultures and Mtb-infected BAL cells were measured using cytometric bead array (CBA) assessments of IFN-γ, TNF, and IL-2 (BD Biosciences 560484). Concentrations of IL-9 and IL-15 were determined by ELISA (Biolegend 434704 and 435104, respectively).

Statistical analysis utilized Wilcoxon matched pairs tests, Mann-Whitney U tests and Kruskal-Wallis tests.

### Software packages
No private code was used in this analysis; all software used was either publicly or commercially available. RNAseq data were processed using STAR (v 2.0.4b) for alignment with Ensembl release 76 top-level assembly. Counts were derived using Subread:featureCount (v1.4.5) and quality assessment was performed using RSeQC (v2.3). The R (v3.4.1)/BioConductor packages EdgeR (v2.20.2) and Limma (v3.34.4) were used to adjust counts for differences in library size. Transcriptomic analysis utilized CompBio (https://www.percayai.com/). GraphPad Prism (v10.0.3) was used for graphs and calculation of statistical significance. Microsoft Excel (v2308) was used for spreadsheets. Blender (v2.93.0) and GIMP (v2.10.22) were used for custom image creation. Heat maps were generated using the R (v4.2.1) package ComplexHeatmap (v2.16.0). BioRender and Adobe Illustrator (v27.8.1) were used for final figure creation.

### Reporting summary
Further information on research design is available in the Nature Portfolio Reporting Summary linked to this article.

## Data availability
The RNAseq data generated in this study have been deposited in the Gene Expression Omnibus database under accession codes GSE224055 (Blood) and GSE223999 (BAL) for all study participants who gave specific consent for genomic data sharing. All other data are available in the article and its Supplementary files or from the corresponding author upon request. Source data are provided with this paper.

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

## Acknowledgements

Funding or these studies was provided by R01 HL111523 (Silver/Hoft), VA Merit Review CX001283 (Silver), SLU VTEU HHSN272201300021I (Hoft). At CWRU, research bronchoscopies were performed in the Dahms Clinical Research Unit, which is a core facility of the Clinical and Translational Science Collaborative (CTSC) of Cleveland funded by UL1TR002548.

## Author contributions

R.F.S., D.F.H.: study design and supervision, analysis, writing; bronchoscopist at CWRU site (RFS); M.X.: RNA extraction, performance of proteomics assays, data analysis; C.E.S., R.D.H.: supervision of RNASeq, extended analysis of genomics data, figure production; J.R.J.: participant recruitment, scheduling, sample processing and immunostaining; M.C.M.: sample processing, immunostaining; A.B.: Lab manager coordinating SLU BAL and Vaccine Center related-work; D.A.S.: bronchoscopist at SLU site; E.K.R.: Nurse Anesthetist Supervisor of SLU bronchoscopy site, assessment and evaluation of participants; JMT: Nurse Coordinator at SLU Vaccine Center Clinic, recruitment of participants, blood draws and coordination of SLU participant research; J.B.G.: transcriptomic analysis of BCG-vaccinated participants' blood responses.

## Competing interests

Rich Head and Chad Storer may receive royalty income based on the CompBio technology developed by them and their collaborators at Washington University, and licensed by Washington University to PercayAi. The remaining authors declare no competing interests.
