## [Peer Review File · Nature Communications]

Distinct gene expression signatures comparing latent tuberculosis infection with different routes of Bacillus Calmette-Guérin vaccinationREVIEWER COMMENTS

Reviewer #1 (Remarks to the Author):

This is a very interesting study comparing three groups of subjects, which are persons with latent TB, and persons who have BCG intradermally (ID), and/or by mouth (PO) – after consent, compared to naive controls.

Analysis includes data derived from lab processed blood and bronchoalveolar lavage cells. The experiments are done pre- and post- ex-vivo stimulation of these cells, with either BCG and the setting of the blood cells or Mtb in the setting of the bronchoalveolar lavage cells. The study seeks to understand an important issue, namely the underperformance of BCG vaccination in preventing pulmonary Mtb infection in adults.

What follows is an extensive amount of bioinformatic analysis of data, (including CompBio). Baseline expression is compared to stimulated data between the four groups, followed by system/pathways analysis. Specific pathways are emphasized, and correlation of gene changes and downstream predicted gene changes is made (e.g. 7B) particularly viz MIT/MR1/xenophagy transcripts. A focused hypothetical construct is offered as ‘one pathway’ - that links PO vaccination (maybe on top of ID vaccination) with macrophage processing of riboflavin followed by MAIT activation with elevated TNF and inflammasome activation.

This is a remarkable and unique data set, and is not accessible elsewhere. The data is descriptive and, as presented generates a significant amount of hypotheses, and the authors offer a hypothetical schema of MAIT cell pathways that may underpin an effect of oral BCG vaccination – but authors might note in their discussion that mechanistic cellular immunology studies will confirm the roles suggested by this bioinformatic screen. Can the author make any comment or reference on the clinical effectiveness of PO vaccination in humans?

IL-9: A conclusion of this paper is that PO BCG vaccination induces mucosal IL-9 production.

While Figure 4 (blood arm) shows significant induction of IL-9 by ID BCG it is not induced by PO BCG, and Figure 6 (BAL arm) does not show IL-9 expression at all – can a graph of IL-9 expression in BAL be included in Figure 6, given the weight this cytokine is given in the subsequent discussion and conclusions of the paper? Also, while Table I does demonstrate significant induction of IL-9 in Mtb-stimulated BAL from the PO±ID BCG group, the levels of IL-9 produced are low (0.0085 is the number, though no concentration unit is given) – so could the authors include some references to support their inference that these IL-9 levels are functionally relevant?

Study Population Groups – a request for more detail please.

This is a remarkable study, that utilizes state of the art technology to characterize four distinct populations (i.e. mycobacterial-naïve, LTBI, ID BCG vaccinated and PO±ID BCG vaccinated), and it offers unique insight into site-specific immune signatures in different settings.

Because the value of its observations hinge upon the careful distinction between populations, if possible, can the authors share a little more detailed description of these groups please? .. offered early in the manuscript.

With regard to the LTBI group, the Methods section explains that these subjects were identified by self-reporting previous LTBI followed by a confirmatory PPD + IGRA, and all reported completing a course of antibiotic therapy. It would be interesting to know LRBI regimen, as waning of this immune signature over time may/may not be an issue.

Description of the BCG vaccinated subjects; For the BAL arm, no explanation for the differing routes of historical vaccination (ID vs PO±ID) for subjects included in the BAL arm is offered – were they all ID? Please say so. Please state that all PO BCG was given for what trial and include the reference for that PO BCG trial published, if this is correct.

With regard to the “PO” BCG vaccinated group in the BAL arm of the study specifically, authors should be clear on how many of these patients had also been vaccinated with ID

BCG, and also within Table I please provide a breakdown of duration since ID vaccination and PO vaccination separately within this “PO±ID” group (it is unclear whether combined vaccination routes were used during an initial vaccination or whether these represent sequential vaccinations). Throughout the manuscript and figures and figure legends, the label “PO” within the BAL arm should be replaced with “PO±ID” to prevent the reader over-interpreting results and inappropriately comparing results between the Blood and BAL arms.

Specific Comments:

Pg 6. Line 178 – why did authors isolate CD4+ T cells for peripheral blood studies but chose to look at mixed-cell BAL rather than isolated T cells for BAL studies? As acknowledged in Methods section (Page 24. Line 586) this limits the interpretation of results from the BAL arm of the study. Obviously it allowed detection of signals in the largely CD4- MAIT cells within the airway, but it would be interesting to know the rationale for this differing approach to begin with.

Pg 6. Line 191 – It is not clear why the authors chose to use a different organism for mycobacterium challenge for the Blood arm (BCG) versus the BAL arm (Mtb), and clarification around why this choice was made would be helpful.

Pg 13. Line 332 – can authors indicate that DDP4 is a marker of activated / Mtb-stimulated MAIT cells rather than just a marker of MAIT cells (as it clarifies why there is an upregulation post ex vivo mycobacterial challenge as demonstrated in Supplementary Figure 4)

Pg 15. Line 385 – please add reference

Pg 15. Line 390 – it is not clear to me how this study demonstrates that “the inadequate respiratory protection following ID BCG is due to limited lung trafficking rather than waning of immunity after years” – all subjects in the blood arm were recently vaccinated, and all of the subjects in the BAL arm were vaccinated years previously, thus waning of immunity could explain this difference?

Reviewer #2 (Remarks to the Author):

This article is the first comprehensive comparison of systemic and pulmonary immunity induced by natural Mtb infection, and by systemic and mucosal BCG vaccination in humans. This makes the data very important. It was carefully performed and has generated very useful data sets. The observation of underlying differences in both blood and BAL between healthy controls and LTBI individuals who have undergone drug treatment is very interesting. The analysis of the BAL and linking this to both the linked blood in LTBI is very important. The data comparisons between LTBI and BCG i.d. and BCG p.o. are interesting but a little confusing. It is this confusion that the figure 8 attempts to address but I think it may overstate.

Main points:

1. This reviewer felt that while the model in fig 8 was intriguing it did not rise to the level of proof required for statements in line 101-104 of the abstract.
2. The study design is good but the RNAseq data could be presented in a better format. Rather than highlighting the differential expression at gene level, it is more usual to consider pathway/module-based differences as these provide for a more robust analysis. In this study a knowledge driven approach was taken and as such the heatmap highlighting gene involving in inflammasome in Figure 7(A) is more compelling than individual gene bar graph in Figure 2b.
3. It will be useful for the reader to provide information on the heterogeneity of gene expression within and between groups. How many differentially expressed genes were identified between groups? Considering the nature and number of the study subjects it is likely that the number of differentially regulated genes are not many. But the information will be useful for the research community as BAL are critical samples and not many RNAseq studies are available. The heterogeneity information will enable the researcher to think what level pathway enrichment can be expected. If there is little difference that is also important information.
4. The authors have mentioned in Figure 7 that the inflammasome signature is unique to the local pulmonary immunity as it is high in the BAL analysis. Is this because of the cell types in each sample i.e. CD4 in blood and 95% macrophage in unsorted BAL. Clarification of the authors thoughts on this is required.
5. 3. Page 6, line 183 “show largely overlapping blood CD4+ T cell gene expression.....”.

Please clarify, “blood CD4+ T cell gene expression” as it does not match with the legend of the figure.

6. Page 7, line 202: Text for figure 2. What is the rationale behind highlighting the 4 genes in figure 2A and 2 genes in figure 2b. This links back to the general topic of using seq data to look at individual genes.

7. page 9, text for Fig 3: A pathway to pathway comparison between the different $\Delta\Delta$ -groups will be more informative for the readers. Please specify in the legend if the size of the nodes denote anything. If it signifies the enrichment score then state whether the range was the same for Fig 3A, 3B and 3C. Similarly, to figure 5 and 7 also.

8. page 10, line 275: targeted protein analysis ($n < 10$) was carried out using multiple systems to assess the level of cytokines in the biofluid. Therefore, the term proteomic can be misleading for the reader.

9. Page 12, line 315: If there is a scope, author could consider doing a western blot to assess protein level caspase-1 and NLRP3 in the stimulated cells.

10. Page 23, please elaborate the method of library preparation. If the same library preparation kit was used for both blood and BAL cells. This aids in future use of the data sets.

11. The authors state that use of antibiotic means that there are no Mtb in the host. This is an assumption. A sentence supporting this assumption should be made.

Reviewer #3 (Remarks to the Author):

Silver and colleagues study the transcriptional response in blood and BAL to various cohorts of individuals related to BCG vaccination and LTBI.

The study represents a really impressive resource to the TB research community as the field seeks to understand how BCG can be utilized more effectively or improved upon.

I believe the study is well-designed and that many of the conclusions may be well supported by the data; however, in the present form, my biggest feedback is that the figures and analyses need to be significantly improved in order to convince the reviewer. Firstly, the CompBio software utilized by the authors is not common, and they present their analyses as

if this is intuitively obvious to the readers. It unfortunately was not for this reviewer. I think the figure density could be significantly reduced to convey a clearer message. Secondly, the $\Delta\Delta$ (delta delta) analysis is not commonly used for the analysis of RNA seq data and the authors need to be more clear about how these analyses were performed. At present, the authors explain the delta delta measurement in Figure 1B, however, in it's current presentation it is not sufficiently clear to understand what the underlying mathematic transformations means for the data. Is this log transformed data? Raw counts? More clarity is needed in the main text and figure and not in the methods.

The authors present proteomic validation of their study as a table. This reviewer believes that the table summary is appropriate for a supplementary table; however, the raw data should be shown. It is not presently clear how much IL-9 was detected in each donor and how this compares to the detection limits of the assays used for IL-9. I understand that the results are significant but there isn't sufficient data transparency for the reviewer to examine the results.

In summary, I believe the results of these studies should be published however significant effort must be made to improve the clarity and presentation of the data so that the audience can understand the results clearly.

Response to Reviewers

We thank the reviewers for their thoughtful comments that have allowed us to improve our manuscript (NCOMMS-22-49178) entitled “Distinct gene expression signatures comparing latent tuberculosis infection with different routes of Bacillus Calmette-Guérin vaccination. We appreciate their assessment of our study as “seek(ing) to understand an important issue, namely the underperformance of BCG vaccination in preventing pulmonary Mtb infection in adults”. We are also pleased by their determination that the manuscript presents “a remarkable and unique data set (that) is not accessible elsewhere” as well as “a really impressive resource to the TB research community as the field seeks to understand how BCG can be utilized more effectively or improved upon”. In our responses to the detailed comments of each reviewer, we have sought to address their questions regarding the subjects and study design, as well as their recommendations for more effective presentation of our findings. Paragraphs that are predominantly new or re-written are now indicated by vertical lines in the left margin whereas changes involving only a few sentences are presented in italics with shadowing.

Reviewer #1:

This is a very interesting study comparing three groups of subjects, which are persons with latent TB, and persons who have BCG intradermally (ID), and/or by mouth (PO) – after consent, compared to naive controls.

Analysis includes data derived from lab processed blood and bronchoalveolar lavage cells. The experiments are done pre- and post- ex-vivo stimulation of these cells, with either BCG and the setting of the blood cells or Mtb in the setting of the bronchoalveolar lavage cells. The study seeks to understand an important issue, namely the underperformance of BCG vaccination in preventing pulmonary Mtb infection in adults.

What follows is an extensive amount of bioinformatic analysis of data, (including CompBio). Baseline expression is compared to stimulated data between the four groups, followed by system/pathways analysis. Specific pathways are emphasized, and correlation of gene changes and downstream predicted gene changes is made (e.g. 7B) particularly viz MIT/MR1/xenophagy transcripts. A focused hypothetical construct is offered as ‘one pathway’ - that links PO vaccination (maybe on top of ID vaccination) with macrophage processing of riboflavin followed by MAIT activation with elevated TNF and inflammasome activation.

This is a remarkable and unique data set, and is not accessible elsewhere. The data is descriptive and, as presented generates a significant amount of hypotheses, and the authors offer a hypothetical schema of MAIT cell pathways that may underpin an effect of oral BCG vaccination – but authors might note in their discussion that mechanistic cellular immunology studies will confirm the roles suggested by this bioinformatic screen. Can the author make any comment or reference on the clinical effectiveness of PO vaccination in humans?

We appreciate Reviewer #1’s supportive comments regarding the study design and our novel findings. As suggested, we have added comments at the end of the Discussion (page 24, lines 568-572) indicating that further functional studies must be performed to confirm the significance of some of the hypotheses generated by the presented analysis generally and with regard to MAIT cells specifically. Unfortunately, there are no publications that report the results of clinical efficacy or effectiveness of PO BCG, as we have now indicated in the Introduction (pages 5-6, lines 167-169). However, there are publications reporting the safety of PO BCG which we already cite in our manuscript. In addition, there are multiple publications focused on pre-clinical models (mice, guinea pigs and nonhuman primates)

that demonstrate PO or other mucosal forms of BCG delivery do protect against Mtb challenges, and we have already cited many of these.

IL-9: A conclusion of this paper is that PO BCG vaccination induces mucosal IL-9 production. While Figure 4 (blood arm) shows significant induction of IL-9 by ID BCG it is not induced by PO BCG, and Figure 6 (BAL arm) does not show IL-9 expression at all – can a graph of IL-9 expression in BAL be included in Figure 6, given the weight this cytokine is given in the subsequent discussion and conclusions of the paper? Also, while Table I does demonstrate significant induction of IL-9 in Mtb-stimulated BAL from the PO±ID BCG group, the levels of IL-9 produced are low (0.0085 is the number, though no concentration unit is given) – so could the authors include some references to support their inference that these IL-9 levels are functionally relevant?

We appreciate the reviewer's interest in our unexpected findings regarding IL9. We apologize for inadvertently omitting the units of cytokine concentrations reported in Table I. Based on the comments of Reviewer 3, these data have now been replaced with graphic demonstrations of cytokine responses at the protein level for each individual subject (new Figure 5); the data previously presented in Table I are now presented in Supplemental Table II. The new figure and updated table both present the units for all measured cytokines in pg/mL. The graphs indicate more clearly that, despite the low levels of IL-9 protein detected following Mtb stimulation of BAL cells from recipients of PO±ID BCG, these findings are consistent and clearly different from the other study groups, resulting in the observed statistical significance. The IL-9 protein levels within the supernatants are quite low, but are interpretable because of the reliability of the standard curve within the Biolegend IL-9 ELISA kit used for these studies. The lack of detection of IL9 gene expression in BAL cells of these same recipients of PO±ID BCG may simply reflect the much lower prevalence of CD4+ T cells generally, and, accordingly, Th9 cells specifically, within the distal airways, as we now note in the Discussion (page 22, lines 523-526). We also point out findings from our further studies of Th9 and IL-9, including their capacity to contain Mtb in both murine and human in vitro infection models, and their ability to protect mice against Mtb aerosol challenge (pages 22, lines 518-529). These findings are the subject of a separate manuscript that is also currently in revision, and is again provided as a supporting document for the current re-submission. Nevertheless, we agree with the reviewer that several caveats must be applied in interpreting the IL-9 findings from BAL studies in recipients of PO±ID reported currently; these are reflected in revisions to the Abstract (page 3, lines 108-109) and Results (pages 12, lines 305-309), as well as in the Discussion as noted above.

Study Population Groups – a request for more detail please...

This is a remarkable study, that utilizes state of the art technology to characterize four distinct populations (i.e. mycobacterial-naïve, LTBI, ID BCG vaccinated and PO±ID BCG vaccinated), and it offers unique insight into site-specific immune signatures in different settings. Because the value of its observations hinge upon the careful distinction between populations, if possible, can the authors share a little more detailed description of these groups please?

With regard to the LTBI group, the Methods section explains that these subjects were identified by self-reporting previous LTBI, and status confirmed with our own PPD skin-testing. All had negative screening for symptoms and for chest x-ray findings consistent with active tuberculosis (pages 25-26, lines 595-599). With further review of our subject information, we noted that we have only limited documentation of self-reported treatment for LTBI and therefore have removed statements indicating that all subjects had completed this antibiotic therapy; implications of this are now included in a revised portion of the discussion (pages 18-19, lines 436-446).

Description of the BCG vaccinated subjects; For the BAL arm, no explanation for the differing routes of

historical vaccination (ID vs PO±ID) for subjects included in the BAL arm is offered – were they all ID? Please say so. Please state that all PO BCG was given for what trial and include the reference for that PO BCG trial published, if this is correct.

All BCG vaccinated subjects (in both the ID or PO alone, and PO±ID groups) were IGRA negative prior to enrollment and remained IGRA negative during the study. All were healthy young adults with no history of BCG vaccination, previous close contact TB exposures or previous known TB infection or disease. All recipients of PO vaccine had participated in clinical trial DMID-01-351, as is now noted in the Methods section (page 25, lines 592-593). For the PO±ID BAL group, all subjects received PO BCG and 2 also received ID BCG on the same day as their PO BCG vaccination. Because we had the smallest PO BCG target population to recruit from, we expanded this group to include together those who either only received PO BCG or received ID BCG on the same day to increase our sample size of volunteers who received at least PO BCG. We similarly combined PO only and PO+ID groups together in our previous Mucosal Immunology report (Reference 14) for the same reason, to optimize the sample size for studies of those that did receive PO BCG. The new Figure 5 mentioned above shows the actual values for cytokine measurements made in blood and BAL supernatants after mycobacterial in vitro stimulation. The 2 PO+ID volunteers are shown by distinct symbols in the BAL graphs. These individualized data clarify that the PO+ID volunteers were not outliers for their group in terms of production of TNF- α and IL-9 expression, for which the PO±ID group responses were significantly and uniquely increased after vaccination. These additional descriptions of the subject groups are included as revisions in Results (page 7, lines 193-196) and Methods (page 25, lines 586-591).

With regard to the “PO” BCG vaccinated group in the BAL arm of the study specifically, authors should be clear on how many of these patients had also been vaccinated with ID BCG, and also within Table I please provide a breakdown of duration since ID vaccination and PO vaccination separately within this “PO±ID” group (it is unclear whether combined vaccination routes were used during an initial vaccination or whether these represent sequential vaccinations). Throughout the manuscript and figures and figure legends, the label “PO” within the BAL arm should be replaced with “PO±ID” to prevent the reader over-interpreting results and inappropriately comparing results between the Blood and BAL arms.

As stated above, among the 8 PO±ID group volunteers, 6 had received PO BCG only and 2 received both PO+ID on the same day, as is now clarified within the text. We have now used the term PO±ID for all references to this group as suggested by the reviewer. Because the two vaccines were given at the same time, however, it was not necessary to add clarification regarding the interval between vaccination and research bronchoscopy to the subject information provided in Supplemental Table I. To provide one readily-visualized comparison of the BAL studies of the six recipients of PO BCG only to those two subjects who received PO+ID, we have indicated their cytokine protein findings separately in the revised Figure 5. As indicated in those graphs, recipients of PO+ID BCG were not outliers in these studies (Legend to Figure 5, page 42, lines 1001-1010)

Specific Comments:

Pg 6. Line 178 – why did authors isolate CD4+ T cells for peripheral blood studies but chose to look at mixed-cell BAL rather than isolated T cells for BAL studies? As acknowledged in Methods section (Page 24. Line 586) this limits the interpretation of results from the BAL arm of the study. Obviously it allowed detection of signals in the largely CD4- MAIT cells within the airway, but it would be interesting to know the rationale for this differing approach to begin with.

We appreciate the reviewers' interest in this issue and the importance of addressing it in the revised manuscript. As we have now clarified in the updated Discussion (pages 16-18, lines 388-420), the differences in approach between blood and BAL studies reflected several factors. Most notably, the

methodologies reflected efforts to generate new data that could be compared with prior studies from each PI's laboratory. Because the Hoft laboratory had extensive experience in banking PBMCs from participants in BCG vaccination trials, we followed their protocols for these assessments. Their earlier choices had been based on many lines of evidence that CD4+ T cells are the primary population that mediates protection against Mtb. The choice of BCG as the in vitro stimulus reflected the origins of this approach in BCG vaccination trials, as the main point was to assess immunity induced by BCG; further, future applications of this approach in wider vaccine trials would be facilitated by use of a non-BSL3 organism. In contrast, as the Silver lab's studies have largely focused on BAL cell responses in individuals with LTBI, Mtb was used as the most relevant antigen. Further, positive selection of CD4+ T cells from BAL would yield extraordinarily limited cell numbers, based on the typical yield of BAL cells in healthy subjects generally, and the low percentage of these that are lymphocytes of any subset. Indeed, Dr. Silver's prior studies directed at assessing the contribution of CD4+ T cells to local gene expression signatures in LTBI was of necessity performed by depleting CD4+ T cells rather than attempting to isolate them. Although the combination of these two approaches makes the findings from blood and BAL cells more difficult to compare directly, the strengths of each approach led to unique findings that might have otherwise been missed. Specifically, enrichment of blood CD4+ T cells was likely essential to identify a role of IL9 in these responses, whereas the implication of CD4- MAIT cells in local immunity to Mtb could not have been detected if BAL studies had been specifically focused on CD4+ T cells alone.

Pg 6. Line 191 – It is not clear why the authors chose to use a different organism for mycobacterium challenge for the Blood arm (BCG) versus the BAL arm (Mtb), and clarification around why this choice was made would be helpful.

As noted above (and now indicated in the text), these approaches reflected the prior experience of each PI's laboratory as well as the populations from which their accumulated previous data has been drawn. This history, as well as potential implications for interpretation of our current findings, is included in the revised Discussion (pages 16-18, lines 388-420).

Pg 13. Line 332 – can authors indicate that DDP4 is a marker of activated / Mtb-stimulated MAIT cells rather than just a marker of MAIT cells (as it clarifies why there is an upregulation post ex vivo mycobacterial challenge as demonstrated in Supplementary Figure 4)

This change has been made in the Results (page 14, lines 353-356) and in the Legend to re-numbered Supplemental Figure 7 (page 50, lines 1176-1182).

Pg 15. Line 385 – please add reference

We believe this line refers back to studies of LTBI in macaque models as well as to meta-analyses of prior studies of LTBI and Mtb exposure that were referenced in the Introduction (page 5, lines 147-151). These references have now been cited again in re-emphasizing this point in the Discussion (page 19, lines 450-452) as well.

Pg 15. Line 390 – it is not clear to me how this study demonstrates that “the inadequate respiratory protection following ID BCG is due to limited lung trafficking rather than waning of immunity after years” – all subjects in the blood arm were recently vaccinated, and all of the subjects in the BAL arm were vaccinated years previously, thus waning of immunity could explain this difference?

The reviewer's point is well-taken and we now include clarification regarding this point. Although it is true that the blood studies were performed immediately after vaccination whereas BAL studies were performed several years later, the interval between vaccination and research bronchoscopy is far less than the clinically relevant interval between neonatal BCG vaccination and the unreliable protection

from pulmonary Mtb infection observed subsequently in adults. More specific to this study, the interval between vaccination in both ID and PO±ID BCG groups and their BAL studies was shorter than the interval between LTBI diagnosis and BAL procedures in naturally-infected individuals. The fact that the interval between ID BCG vaccination and BAL procedures was shorter than the interval between LTBI diagnosis/treatment and BAL further supports the conclusion that the lower lung responses after ID BCG were unlikely due to time-related waning of immunity alone. However, we agree that our findings have not definitively “demonstrated” that “...inadequate respiratory protection following ID BCG is due to limited lung trafficking rather than waning of immunity...” and have modified the discussion to avoid this possible over-interpretation of the results (pages 19-20, lines 456-466)

Reviewer #2 (Remarks to the Author):

This article is the first comprehensive comparison of systemic and pulmonary immunity induced by natural Mtb infection, and by systemic and mucosal BCG vaccination in humans. This makes the data very important. It was carefully performed and has generated very useful data sets. The observation of underlying differences in both blood and BAL between healthy controls and LTBI individuals who have undergone drug treatment is very interesting. The analysis of the BAL and linking this to both the linked blood in LTBI is very important.

We appreciate the reviewer’s comments regarding the value of the data obtained in this study.

The data comparisons between LTBI and BCG i.d. and BCG p.o. are interesting but a little confusing. It is this confusion that the figure 8 attempts to address but I think it may overstate.

Main points:

1. This reviewer felt that while the model in fig 8 was intriguing it did not rise to the level of proof required for statements in line 101-104 of the abstract.

We agree and have changed the wording of this portion of the abstract to indicate that this concept is suggested rather than proven by the current data (page 3, lines 105-107).

2. The study design is good but the RNAseq data could be presented in a better format. Rather than highlighting the differential expression at gene level, it is more usual to consider pathway/module-based differences as these provide for a more robust analysis. In this study a knowledge driven approach was taken and as such the heatmap highlighting gene involving in inflammasome in Figure 7(A) is more compelling than individual gene bar graph in Figure 2b.

We appreciate the reviewer’s advice and have substantially revised our data presentation to address these concerns. Most of the individual-gene bar graphs within the main figures have been replaced with heat maps of key biologically-grouped genes sets from the knowledge-driven analysis (eg, current Figures 3 and 4). The remaining bar graphs in the main figure set (see revised Figure 7b) were retained as we felt they demonstrate an important point relevant for the MAIT cell activation hypothesis with the elevated baseline expression levels of two of the key genes specifically within the BAL PO±ID group. We continue to provide representative single-gene bar graphs in new Supplemental Figures 2, 4, and 6 (and re-numbered Supplemental Figure 7) to provide additional granularity for the results.

3. It will be useful for the reader to provide information on the heterogeneity of gene expression within and between groups. How many differentially expressed genes were identified between groups? Considering the nature and number of the study subjects it is likely that the number of differentially regulated genes are not many. But the information will be useful for the research community as BAL are critical samples and not many RNAseq studies are available. The heterogeneity information will enable the researcher to think what level pathway enrichment can be expected. If there is little difference that is also important information.

We agree that providing the numbers of differentially-expressed genes in the various comparisons is helpful to the reader in evaluating the data and resulting conclusions. This information is now provided for Δ and $\Delta\Delta$ comparisons in studies of blood and BAL within the four components of new Supplemental Table III.

4. The authors have mentioned in Figure 7 that the inflammasome signature is unique to the local pulmonary immunity as it is high in the BAL analysis. Is this because of the cell types in each sample i.e. CD4 in blood and 95% macrophage in unsorted BAL. Clarification of the authors thoughts on this is required.

The reviewer's point is well-taken. As we have now stated more explicitly (Discussion, pages 17-18, lines 405-420, and page 21, lines 502-505), the putative connection of MAIT cells to the inflammasome signature in BAL cells not detected in our blood studies, could be due both to high proportions of macrophages in BAL and to exclusion of largely CD4- MAIT cells from the purified CD4/MDDC co-cultures established from peripheral blood. On the other hand, it is also possible that this difference would have been observed even with unselected circulating T cell populations given the greater predominance of MAIT cells at mucosal sites generally.

5. 3. Page 6, line 183 "show largely overlapping blood CD4+ T cell gene expression.....". Please clarify, "blood CD4+ T cell gene expression" as it does not match with the legend of the figure.

Thank you for pointing out this discrepancy. The Results section text and the legend to Supplemental Figure 1 have now been rewritten to clarify that the findings represent gene expression of co-cultured CD4+ T cells and autologous MDDC in the Results section (page 7, lines 186-191), and the Legend to Supplemental Figure 1 (page 47, lines 1111-1120).

6. Page 7, line 202: Text for figure 2. What is the rationale behind highlighting the 4 genes in figure 2A and 2 genes in figure 2b. This links back to the general topic of using seq data to look at individual genes.

The genes presented in bar graphs were selected as exemplars to illustrate different general patterns of expression observed across the various subject groups. However, we appreciate the reviewer's point that further presentation of expression of multiple genes within knowledge-based, immune-associated themes provides greater insight into the findings and the manner in which they were analyzed. These have been included in revised Figures 3, 4, 6 and 7, whereas bar graphs displaying expression of individual genes have been moved into Supplemental Figures 2, 4, and 6 (in addition to data for DPP4/CD26, which remains in supplemental material, now as Supplemental Figure 7).

7. page 9, text for Fig 3: A pathway to pathway comparison between the different $\Delta\Delta$ -groups will be more informative for the readers. Please specify in the legend if the size of the nodes denote anything. If it signifies the enrichment score then state whether the range was the same for Fig 3A, 3B and 3C. Similarly, to figure 5 and 7 also.

As noted above, we have replaced these single-gene assessments with presentations showing expression of multiple genes within representative immune themes/pathways. We also have provided more explanation of the statistical approaches that underlie CompBio analysis, as well as more description of the meaning sizes of CompBio spheres (which correlate with rank of each theme within the analysis), and thickness of connecting edges (based on the number of shared genes). These additions are now provided in the revised Methods section (pages 29-30, lines 686-703), at the point of their first appearance in Figure 3 in the Results (page 9, lines 226-227) and its associated Figure Legend (pages 38-39, lines 929-938)

8. page 10, line 275: *targeted protein analysis (n<10) was carried out using multiple systems to assess the level of cytokines in the biofluid. Therefore, the term proteomic can be misleading for the reader.*

We agree and have removed the term “proteomics” from the description and presentation of these data.

9. Page 12, line 315: *If there is a scope, author could consider doing a western blot to assess protein level caspase-1 and NLRP3 in the stimulated cells.*

We appreciate this suggestion. As is now indicated in the Discussion (page 24, lines, 569-573) this genomic-based study was intended to be exploratory and will ultimately guide the use of other methodologies to confirm the significance of these findings in protection against Mtb. Unfortunately, there were not adequate cells remaining from the work we report here for this type of confirmation, which would in any case be beyond the scope of this project.

10. Page 23, please elaborate the method of library preparation. *If the same library preparation kit was used for both blood and BAL cells. This aids in future use of the data sets.*

We appreciate the reviewer’s point. The specifics for the two different library prep protocols are provided in the revised Methods section (pages 27-28, lines 640-650). The blood-derived samples utilized the Ribo-ZERO kit (Illumina-EpiCentre) and the BAL samples, processed later, required use of the SMARTer Ultra Low RNA kit (Takara-Clontech).

11. *The authors state that use of antibiotic means that there are no Mtb in the host. This is an assumption. A sentence supporting this assumption should be made.*

We thank the reviewer for emphasizing this point, and have re-written this aspect of the Discussion substantially (pages 18-19, lines 436-447). As noted in the response to Reviewer #1, we now acknowledge that we do not have confirmation of the LTBI subjects’ self-reported completion of therapy, and are aware the non-compliance rates with LTBI treatment are substantial. Further, there is no reliable assay for confirming that the goal of complete sterilization of viable Mtb has been successfully achieved following treatment of a given individual with LTBI. We have therefore re-written this component of the discussion to acknowledge that the continued presence of viable Mtb cannot be excluded in treated LTBI subjects, but also note that we do not detect increased baseline cytokine production in blood or BAL cells of LTBI subjects. Therefore, other mechanisms of ongoing immune activation, such as epigenetic macrophage conditioning or reprogramming, may provide alternative explanations for these findings and should be explored in subsequent studies.

Reviewer #3 (Remarks to the Author):

Silver and colleagues study the transcriptional response in blood and BAL to various cohorts of individuals related to BCG vaccination and LTBI.

The study represents a really impressive resource to the TB research community as the field seeks to understand how BCG can be utilized more effectively or improved upon. I believe the study is well-designed and that many of the conclusions may be well supported by the data;

We thank the reviewer for this appreciation of the value of our study findings.

... in the present form, my biggest feedback is that the figures and analyses need to be significantly improved in order to convince the reviewer. Firstly, the CompBio software utilized by the authors is not common, and they present their analyses as if this is intuitively obvious to the readers. It unfortunately

was not for this reviewer. I think the figure density could be significantly reduced to convey a clearer message.

We appreciate the Reviewer's perspective on our presentation of study methods and results. As noted in responses to Reviewers 1 and 2, we now provide a more detailed overview of the type of analysis utilized in the CompBio assessments (pages 28-31, lines 666-716), as well as references for more detailed explanation of this approach (References 42-44). Further, in the revised Figures 3, 4, 6 and 7, we present data for specific immune themes/pathways identified as relevant to the comparisons between subject groups in terms of more focused heat maps to demonstrate how the responses of specific functionally-associated genes contribute to these conclusions.

Secondly, the $\Delta\Delta$ (delta delta) analysis is not commonly used for the analysis of RNA seq data and the authors need to be more clear about how these analyses were performed. At present, the authors explain the delta delta measurement in Figure 1B, however, in its current presentation it is not sufficiently clear to understand what the underlying mathematic transformations means for the data. Is this log transformed data? Raw counts? More clarity is needed in the main text and figure and not in the methods.

Naïve and mycobacteria immune persons all respond strongly to BCG/Mtb in vitro so that $\Delta\Delta$ comparisons allow for a better dissection of unique DEGs associated with antigen-specific memory responses. However, we now further emphasize that not all assessments are based on $\Delta\Delta$ comparisons. Specifically, the high baseline immune activation in blood and BAL of LTBI subjects is again emphasized (Results, pages 7-8, lines 199-221; Discussion, pages 18-19, lines 418-436, and Table 1). We also now indicate that, with regard to our main findings, $\Delta\Delta$ comparisons may tend to make infection-induced responses of LTBI subjects appear less impressive than those of other study groups, as is now included as a caveat within the Discussion (page 18, lines 428-435)

The authors present proteomic validation of their study as a table. This reviewer believes that the table summary is appropriate for a supplementary table; however, the raw data should be shown. It is not presently clear how much IL-9 was detected in each donor and how this compares to the detection limits of the assays used for IL-9. I understand that the results are significant but there isn't sufficient data transparency for the reviewer to examine the results.

We appreciate the reviewer's interest in these raw data. Graphs of cytokine protein concentrations in cultured samples including the data points of individual subjects are now included as Figure 5 in the revised manuscript and reported in the Results section (pages 11-12, lines 280 -309). As described above in response to reviewer #1, the new Figure and updated Table (now Supplemental Table II) specifically indicate that all results are reported as pg/mL. The graphic presentation of these results does show that in BAL samples, the IL-9 levels in recipients of PO±ID BCG, although quite low, were both unique and consistent (as well as statistically significant) in showing Mtb-induced increases. Further discussion of these findings was addressed in responses to Reviewer #1 above.

In summary, I believe the results of these studies should be published however significant effort must be made to improve the clarity and presentation of the data so that the audience can understand the results clearly.

We thank the reviewer for his/her appreciation of the value of our findings and for the recommendations that have helped us to substantially improve this manuscript.

REVIEWERS' COMMENTS

Reviewer #1 (Remarks to the Author):

The author have paid attention to all points made by all reviewers. Their detailed response has been carried over to a number of significant clarifications and changes to the original manuscript. All of the points which we have made and all of the questions we have generated - have been addressed to our satisfaction.

Reviewer #2 (Remarks to the Author):

The authors have addressed my concerns

Reviewer #3 (Remarks to the Author):

The authors have responded to my concerns.